# OUTCOME-BASED EXPLORATION FOR LLM REASONING

## ABSTRACT

Reinforcement learning (RL) has emerged as a powerful method for improving the reasoning abilities of large language models (LLMs). Outcome-based RL, which rewards policies solely for the correctness of the final answer, yields substantial accuracy gains but also induces a systematic loss in generation diversity. This collapse undermines real-world performance, where diversity is critical for test-time scaling. We analyze this phenomenon by viewing RL post-training as a sampling process and show that, strikingly, RL can reduce effective diversity even on the *training* set relative to the base model. Our study highlights two central findings: (i) a *transfer of diversity degradation*, where reduced diversity on solved problems propagates to unsolved ones, and (ii) the *tractability of the outcome space*, since reasoning tasks admit only a limited set of distinct answers. Motivated by these insights, we propose *outcome-based exploration* (`OBE`), which assigns exploration bonuses according to final outcomes. We introduce two complementary algorithms: *historical exploration*, which encourages rarely observed answers via UCB-style bonuses, and *batch exploration*, which penalizes within-batch repetition to promote test-time diversity. Experiments on standard competition math with `Llama` and `Qwen` models demonstrate that both methods improve accuracy while mitigating diversity collapse. On the theoretical side, we formalize the benefit of outcome-based exploration through a new model of *outcome-based bandits*. Together, these contributions provide a practical path toward RL methods that enhance reasoning without sacrificing the diversity essential for scalable deployment.

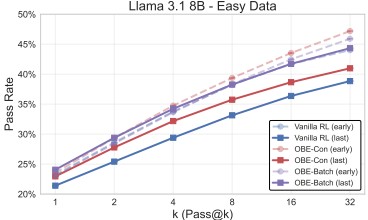 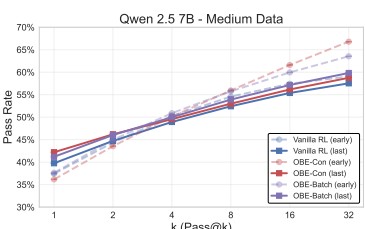

Figure 1: Test performance comparison (averaged across `MATH-500`, `AIME24/25`, `AMC23`) between our exploration methods (`OBE-Con` and `OBE-Batch`) and the `GRPO` baseline, with `Llama-3.1-8B-Instruct` on the easy dataset (left) and `Qwen-2.5-7B-Base` on the medium dataset (right). We report pass@$k$ for $k \in \{1, 2, 4, 8, 16, 32\}$ on an early checkpoint (at timestep 100) and the final checkpoint (at timestep 700). We repeat each experiment with 3 different random seeds and plot the mean performance. The exploration methods outperform the baseline on nearly all metrics across the training process (except `Qwen-2.5-7B-Base` with `OBE-Con` on pass@1 on the early checkpoint due to exploration, but it has much higher pass@32 rate), and better exploitation-exploration trade-off and mitigation of overoptimization (note that the last checkpoint of `Llama-3.1-8B-Instruct` with Vanilla RL has overall worse performance than its early checkpoint due to overoptimization).

## 1 INTRODUCTION

Large language models (LLMs) are commonly post-trained with reinforcement learning (RL), both in preference alignment (Ouyang et al., 2022; Bai et al., 2022) and in reasoning (Shao et al., 2024; Guo

et al., 2025). A longstanding difficulty in RL is the design of reward signals: while one might hope to shape intermediate reasoning steps, recent works have shown that the seemingly crude strategy of rewarding only the final correctness (e.g., whether a math answer is correct) can be remarkably effective (Shao et al., 2024; Guo et al., 2025). However, a growing body of evidence points to an important drawback of RL post-training: a systematic loss of diversity in model generations (Song et al., 2024b; Dang et al., 2025; Yue et al., 2025; Zhao et al., 2025; Wu et al., 2025). This phenomenon is most cleanly captured by the pass@$k$ metric: when $k$ is large (e.g., $k = 512$), post-trained models exhibit a lower pass@$k$ than the base model, usually on in-distribution validation sets. This raises a practical concern: in real-world deployments, diversity is often valuable and can amplify performance through test-time scaling (Wu et al., 2024; Snell et al., 2024), with different sampling processes such as directly sampling from the model or tree search. Indeed, we find that diversity degradation already manifests during training, as models collapse to a reduced set of candidate answers on unsolved problems due to a transfer effect of the diversity degradation induced by concentrating on correct answers, which we detail in Section 2.

Exploration is the canonical RL tool for combating such collapse (Bellemare et al., 2016; Azar et al., 2017; Burda et al., 2018). However, directly importing classical techniques such as Upper Confidence Bound (UCB) exploration (Auer et al., 2002) to token-level language modeling is intractable, as it would require searching over exponentially many sequences. Motivated by the success of outcome-based rewards, we therefore study *outcome-based exploration* (OBE), where exploration bonuses depend only on final outcomes. This perspective allows us to adapt UCB-style methods to LLM training, which we further refine by incorporating both positive and negative outcome signals.

A subtlety arises, however: in language models, one must distinguish between *historical exploration* (visiting a more diverse set of states and actions during training) and *batch exploration* (producing diverse outputs at test time). The latter improves pass@$k$ but does not necessarily increase diversity during training whereas the former improves pass@1 but does not guarantee test-time diversity of the trained model. We introduce and study a batch version of outcome-based exploration, which demonstrates improved tradeoff between accuracy and diversity during test time. Our contributions are as follows:

1. We study RL post-training dynamics by framing RL as a sampling process (Section 2). This perspective reveals that diversity loss is not limited to test-time behavior, but already occurs on training: as RL concentrates probability mass on solved questions, the resulting collapse propagates and reduces diversity even on unsolved ones. We term this effect the transfer of diversity degradation.

2. We propose outcome-based exploration (Section 3), which adapts classical exploration bonuses (e.g. UCB) to the outcome space of LLM tasks. We show that naively adapting UCB does not lead to improved testing performance. We thus propose more refined algorithms (OBE-Mean, OBE-Con) which incorporate both positive and negative signals, and show that they improve both training exploration and test generalization, on standard reasoning dataset such as DAPO and models such as Llama-3.1-8B-Instruct. We further provide a theoretical analysis on the benefit of outcome-based exploration in a new bandit setting (outcome-based bandits), inspired by the practical considerations (Section 4.2).

3. We introduce a batch version of the OBE algorithm (OBE-Batch) (Section 3.3). By penalizing repetitive answers within the latest samples, the algorithm explicitly encourages diverse generations on the batch level, yielding a better accuracy–diversity tradeoff at test time.

4. We analyze the interaction between historical and batch exploration, showing that they are not mutually exclusive (Section 4.1). In summary, our proposed methods can be easily incorporated into standard RL for LLMs reasoning training, agnostic to the training algorithm, and consistently improve both accuracy and diversity. Our algorithms are also supported by scientific study and theoretical analysis, which alone provides an independent interest to the community.

## 2 DIVERSITY DEGRADATION: RL AS SAMPLING

**Preliminaries**   We consider LLM reasoning training with RL in a verifiable reward setting. Denote the set of questions as $\mathcal{X}$, the training question set $\mathcal{X}_{\text{train}} \subseteq \mathcal{X}$ and the test question set $\mathcal{X}_{\text{test}} \subseteq \mathcal{X}$. Further, define the space of intermediate text as $\mathcal{Y}$, and the answer space as $\mathcal{A}$; we consider an LLM to be a policy $\pi : \mathcal{X} \to \Delta(\mathcal{Y} \times \mathcal{A})$, i.e, given any question $x \in \mathcal{X}$, the LLM generates a sample

$(y, a) \sim \pi(\cdot \mid x)$, where $y \sim \pi(\cdot \mid x)$ is the intermediate reasoning trace (chain of thought) and $a \sim \pi(\cdot \mid x, y)$ is the final answer. By following the convention in Guo et al. (2025) we have access to a ground truth reward $r : \mathcal{X} \times \mathcal{A} \to \{0, 1\}$ that checks the correctness of the final answer. The evaluation metric for a given (dataset, policy) pair is defined in terms of the accuracy of the final answer: $J(\pi, \mathcal{X}) = \mathbb{E}_{x \sim \text{unif}(\mathcal{X})} \mathbb{E}_{(y,a) \sim \pi(\cdot \mid x)}[r(x, a)]$. During RL training, we use the KL-regularized version of the objective $J(\pi, \mathcal{X}_{\text{train}})$, which aims to find the $\pi^\star$ such that

$$\pi^\star := \arg\max_\pi \mathbb{E}_{x \sim \text{unif}(\mathcal{X}_{\text{train}})} \big[ \mathbb{E}_{(y,a) \sim \pi(\cdot \mid x)}[r(x, a)] - \beta \text{KL}(\pi(\cdot \mid x), \pi_{\text{base}}(\cdot \mid x)) \big],$$

where $\pi_{\text{base}}$ is our base LLM from which the RL training is initialized. In this paper, we consider the fully on-policy GRPO algorithm (Shao et al., 2024), which optimizes the following objective:

$$\widehat{\mathbb{E}}_{x, \{y_i, a_i\}_{i=1}^n \sim \pi(\cdot \mid x)} \left[ \frac{1}{n} \sum_{i=1}^n \hat{A}(x, \{y_i, a_i\}_{i=1}^n)_i - \beta \widehat{\text{KL}}(\pi(\cdot \mid x), \pi_{\text{base}}(\cdot \mid x)) \right], \tag{1}$$

where $\hat{A}(x, \{y_i, a_i\}_{i=1}^n)_i = \frac{r(x, a_i) - \mu(\{r(x, a_{i'})\}_{i'=1}^n)}{\sigma(\{r(x, a_{i'})\}_{i'=1}^n)}$ and $\widehat{\text{KL}}(\pi(\cdot \mid x), \pi_{\text{base}}(\cdot \mid x))$ is estimated by $\log\left(\frac{\pi(y, a \mid x)}{\pi_{\text{base}}(y, a \mid x)}\right) + \frac{\pi_{\text{base}}(y, a \mid x)}{\pi(y, a \mid x)} - 1$, which is considered to enjoy lower variance than directly sampling $\log\left(\frac{\pi(y, a \mid x)}{\pi_{\text{base}}(y, a \mid x)}\right)$ (Schulman, 2020). Notice that it is known that this objective leads to a biased gradient of the regularized objective $J$, and incidentally minimizes the forward KL-divergence $\text{KL}(\pi_{\text{base}}(\cdot \mid x), \pi(\cdot \mid x))$. (Tang & Munos, 2025). However, we will use this algorithm as it is a popular baseline and will call it vanilla RL in the following.

Finally, given question $x$, we define all possible reasoning traces of LLM $\pi$ as $\mathcal{Y}^\pi(x) = \text{supp}(\pi(\cdot \mid x))$, and thus the answer support of an LLM $\pi$ as $\mathcal{A}^\pi(x) := \text{supp}(\pi(\cdot \mid x, \mathcal{Y}^\pi(x)))$.

**Experiment setting** We focus on LLM RL training for math reasoning. We test two models: `Llama-3.1-8B-Instruct` (Dubey et al., 2024) and `Qwen-2.5-7B-Base` models (Yang et al., 2024). We use two datasets: an easy dataset and a medium difficulty dataset. The easy dataset is the train split of the `MATH` dataset (Hendrycks et al., 2021) with a total of 7500 questions. For the medium dataset, we subsample 3840 questions from the training set of `DAPO` (Yu et al., 2025). We also include an additional hard dataset, where we keep the questions from the easy and medium dataset on which the base model's pass@512 is 0. We defer details on the hard dataset to Section E. To test, we use the `MATH-500`, (Lightman et al., 2023), `AIME24/25`, and `AMC23` datasets (dataset adopted from Shafayat et al. (2025)). To measure whether two given answers are different, we apply the math_verify function in the verl (Sheng et al., 2024) codebase, which treats two answers as the same as long as they are mathematically equivalent. This also defines our reward function since it is the indicator function of whether a given answer is equivalent to the ground truth answer. Implementation details can be found in Section H.

## 2.1 DIVERSITY DEGRADATION DURING RL TRAINING

Recently it has been observed that, during LLM post training (either with SFT or RL), the diversity of the final policy decreases, as measured with the pass@$k$ metric with $k > 1$, over the test dataset (Song et al., 2024b; Dang et al., 2025; Yue et al., 2025; Wu et al., 2025). However, the previous analysis only focused on comparing the base model $\pi_{\text{base}}$ with the final model checkpoint $\pi_T$, a single artifact of the RL training method.

To understand how diversity degrades during RL training, we propose to examine the dynamics of RL training by considering RL as a sampling process on the training set. Specifically, in each epoch $t \in [T]$ during RL training, one samples $n$ trajectories for each question $x \in \mathcal{X}_{\text{train}}$. Thus given a base model $\pi_{\text{base}}$ and RL algorithm (denoted as Alg), we sample in total $nT$ trajectories for each question $x$, i.e., $\{y_i, a_i\}_{i=1}^{nT} \sim \text{Alg}(\pi_{\text{base}}, x)$. Now this allows us to directly compare with sampling the same amount of trajectories from the base model, i.e., $\{y_i', a_i'\}_{i=1}^k \sim \pi_{\text{base}}(\mathcal{X}_{\text{train}})$, where $k = nT$.

We conducted experiments on the `Llama-3.1-8B-Instruct` and `Qwen-2.5-7B-Base` models, trained on both the easy and medium difficulty datasets. To compare the RL training dynamics and the base model, we adopt two metrics: total number of questions solved and total number of distinct answers. Note that these metrics correspond to the pass@$k$ and diff@$k$ metrics that are used to measure a fixed model. Recall that to convert a training epoch $t$ to $k$ in pass@$k$ and diff@$k$, we have $k = nt$.

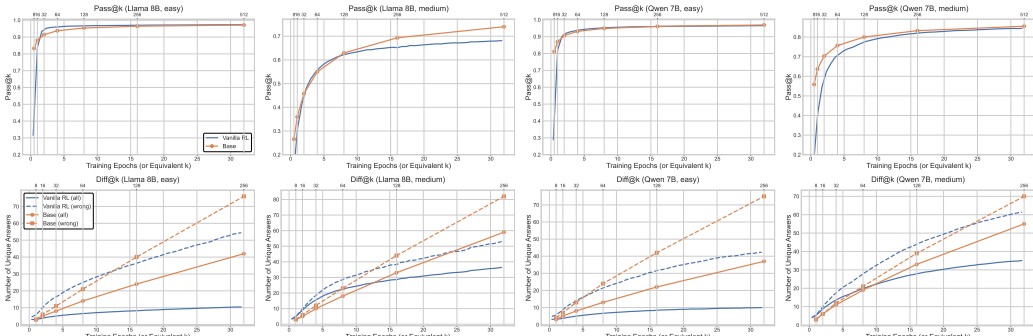

Figure 2: Comparison between RL training dynamics and base model sampling, on both easy and medium difficulty datasets, with `Llama-3.1-8B-Instruct` and `Qwen-2.5-7B-Base`. Top row: number of questions solved so far; Bottom row: number of different answers per question sampled so far. The bottom x-ticks are the number of epochs $t$ for training, and the top x-ticks are the corresponding $k$ for sampling from the base model. We convert $k = nt$ where $n$ is the number of samples per group and $t$ is the epoch index. We use $n = 16$ for pass@$k$ comparison and $n = 8$ for diff@$k$ comparison. In the diff@$k$ comparison, solid lines denote the average number of different answers per all questions, and dashed lines denote the average number of different answers per unsolved questions (i.e., all answers are wrong so far). The fact that RL has lower diff@$k$ on unsolved questions than the base model (dashed lines) indicates the transfer of diversity degradation.

For all of our experiments, we use standard hyperparameters recorded in Section H unless otherwise specified. We summarize the results in Figure 2, and we make the following observations:

- **RL eventually solves fewer questions than the base model**. At the beginning of the RL training, the rate of questions solving is faster than the base model, which is expected, as RL quickly converges to the correct answers on the ones that it can easily solve. However, as training continues, the rate of question solving decreases faster than the base model, and eventually RL solves fewer questions than the base model with the same amount of samples *on the training set*.

- **Transfer of diversity degradation.** In an ideal setting where the training dynamics are independent across questions, vanilla RL training should never underperform the base model. This is because the model does not update on questions $x$ it has not solved yet (i.e., it receives zero gradient on those questions), so its behavior on those questions is equivalent to the base model, i.e., $\mathcal{A}^{\pi_{\text{RL}}}(x) = \mathcal{A}^{\pi_{\text{base}}}(x)$. The observed diversity degradation can therefore be explained as follows: once the model concentrates its answers on questions it has solved, this reduced diversity propagates to unsolved questions as well. To quantify this effect, we track the cumulative number of distinct answers sampled. We find that RL training yields lower diversity across all questions on average, and, more importantly, even lower diversity on the unsolved questions. We refer to this phenomenon as the *transfer of diversity degradation*.

- **Diversity is tractable on verifiable domains**. In general it is hard to predict that, given two generations from LLMs, whether they are semantically different or not. Naively measuring in token space results in exponentially many candidates and thus is intractable. However, in the verifiable domain, we can use the final answer as a proxy to measure the diversity of the generations. From Figure 2, we observe that given a large sample budget, we only have $|\mathcal{A}^{\pi_{\text{base}}}(x)| < 50$ on average, which is tractable to measure and optimize. We refer to this property as *the tractability of the outcome space*. We will introduce our algorithms that leverage this property in the next section.

## 3 OUTCOME-BASED EXPLORATION

### 3.1 HISTORICAL EXPLORATION

Given the observation that there are bounded number of final answers to search over, our training objective thus becomes to explore as many different answers (and their corresponding reasoning traces) as possible, while also rewarding the correctness of the answer. This problem is well studied

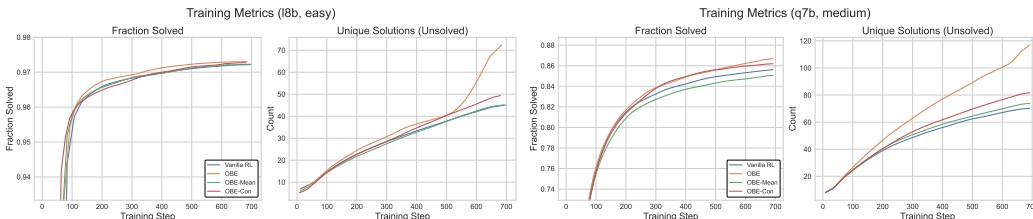

Figure 3: Training performance comparison between different `OBE` variants and the `GRPO` baseline, with `Llama-3.1-8B-Instruct` on the easy dataset (left) and `Qwen-2.5-7B-Base` on the medium dataset (right). For each subplot: Left: fraction of questions solved so far; Right: number of different answers sampled on the questions that the model has yet to solve (i.e., sample one correct answer historically). The x-axis denotes the number of gradient updates as we train all models fully on-policy. We repeat each experiment with 3 different random seeds and plot the mean performance.

in the bandit and RL literature, and the canonical solution for exploration is the upper confidence bound (UCB) method (Auer et al., 2002; Azar et al., 2017), which for each state and action adds an additional exploration bonus that is inversely proportional to its historical visitation counts, on top of the correctness reward. Thus, the training objective in Eq. (1) becomes:

$$\widehat{\mathbb{E}}_{x,\{y_i,a_i\}_{i=1}^n \sim \pi(\cdot|x)} \left[ \frac{1}{n} \sum_{i=1}^n \widehat{A}(x, \{y_i, a_i\}_{i=1}^n)_i + c \cdot b(x, a_i) - \beta \widehat{\mathrm{KL}}(\pi(\cdot \mid x), \pi_{\mathsf{base}}(\cdot \mid x)) \right], \quad (2)$$

where $c$ is a tunable hyperparameter and the bonus term resembles the `UCB` bonus

$$b(x, a) = \min\left\{ 1, \sqrt{\frac{1}{N(x, a)}} \right\},$$

where $N(x, a)$ is the number of times we have sampled the answer $a$ for the question $x$. Note that the exact form of `UCB` bonus contains other terms such as the number of arms and rounds. Since these terms are constants, we absorb them in the coefficient $c$. We denote this method as `OBE`. In practice, to mitigate bonus hacking, we propose to further mask the final answer during the policy update. In this case, the gradient will only go through $\pi(y \mid x)$ but not $\pi(a \mid y, x)$.

### 3.1.1 NAIVE `OBE` ONLY IMPROVES TRAINING PERFORMANCE

We present training results in Figs. 3 and 13 and test results in Figs. 4 and 14. We observe that, although `OBE` improves the training performance consistently (and with a larger improvement on the harder dataset), it does not consistently improve the test performance across all models and datasets. In particular, we observe a significant improvement only on the easy dataset with the `Llama-3.1-8B-Instruct` model, but not in the other settings.

Originally, the construction of the `UCB` bonus is due to the stochasticity of the problem. For any pair of state and action $(x, a)$, the estimation error of the dynamics and reward scales with the order of $O(1/\sqrt{N(x, a)})$, and the bonus offsets this error and encourages the policy to explore underexplored states and actions. On the contrary, in the LLM reasoning setting, the dynamics and reward are both deterministic, and thus in the extreme case where the training dynamics are independent across questions, the policy should stop visiting an answer once it gets a reward of 0, because now it has a perfect estimation of the reward of this answer. However, in practice, the training dynamics is not independent across questions (cf. Section 2.1), and intuitively the bonus encourages the model to explore answers that it has not visited often and thus accelerates the training performance. However, only providing positive bonuses might still lead to over-visitation of wrong answers, and we hypothesize that this overfitting issue hurts the generalization performance.

### 3.2 `OBE` WITH A BASELINE

The above observation suggests that providing only positive exploration signals is not the most effective strategy where test performance is concerned. Instead, we propose incorporating a baseline

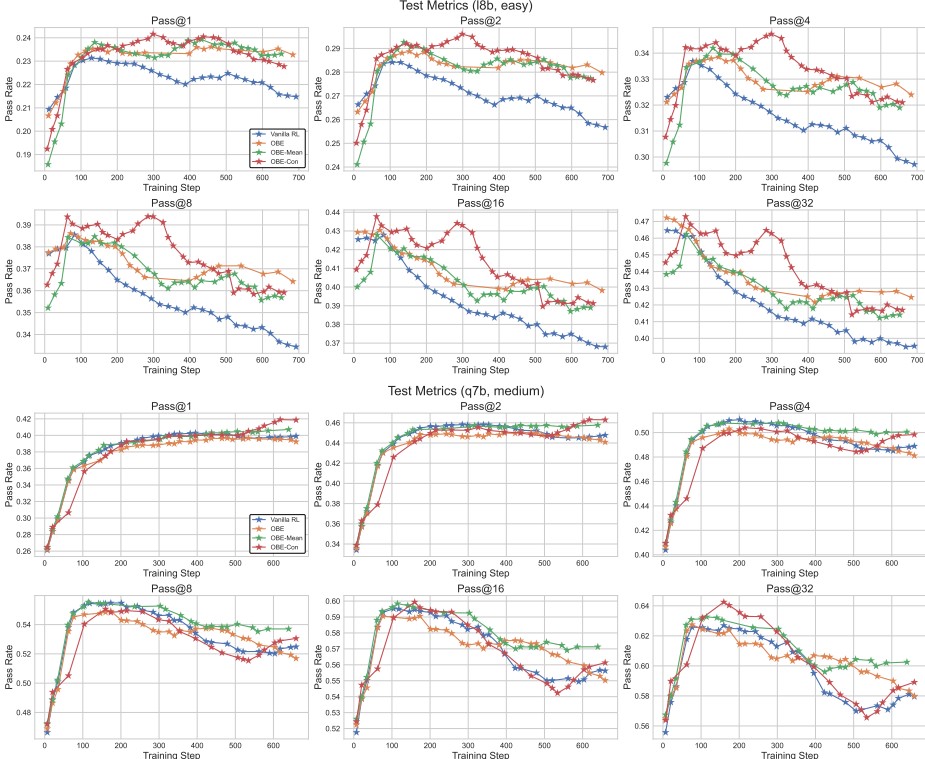

Figure 4: Test performance comparison between different `OBE` variants and the `GRPO` baseline, with `Llama-3.1-8B-Instruct` on the easy dataset (top) and `Qwen-2.5-7B-Base` on the medium dataset (bottom). We report pass@$k$ for $k \in \{1, 2, 4, 8, 16, 32\}$ at every 20 training steps. We repeat each experiment with 3 different random seeds and plot the mean performance (see Section G for error bars). The metrics are calculated based on 32 samples per question during evaluation.

into the bonus calculation, so that exploration signals are defined relative to this baseline and can be either positive or negative. A natural starting point—analogous to the `GRPO` baseline—is to use the batch mean of the `UCB` bonus as the baseline. Concretely, we modify the objective in Eq. (2) by replacing $b(x, a_i)$ with:

$$\widehat{B}(x, \{y_i, a_i\}_{i=1}^n)_i = b(x, a_i) - \frac{1}{n} \sum_{j}^n b(x, a_j).$$

We refer to this method as `OBE` with a mean baseline (`OBE-Mean`). Intuitively, it encourages the model to explore answers that are less frequent in the current batch while penalizing those that appear more often. Although historically frequent answers tend to receive a negative signal, since the baseline is calculated on the batch level, a well-visited answer can still receive a positive exploration signal if it is relatively underrepresented within the current batch. To avoid this issue, we propose a third method, `OBE` with a constant baseline (`OBE-Con`), where we simply use a constant as the baseline, i.e.,

$$\widehat{B}(x, \{y_i, a_i\}_{i=1}^n)_i = b(x, a_i) - b_0,$$

where $b_0$ is a tunable hyperparameter. Note that this gives easy control over the tradeoff between positive and negative exploration signal (Arnal et al., 2025), even though in expectation the gradient of the baseline is 0. For example, if we set $b_0 = 0.5$, then an answer will get a positive exploration signal if it has been visited less than 4 times, and a negative signal otherwise. One issue with the baseline formulation is that, in the case where all answers in the batch are correct, then we have $A_i = 0$ for all $i$, and thus the exploration signal will dominate the training objective. After the beginning of the training, this objective will assign a negative gradient to the batch where all the answers are correct. To prevent this undesirable behavior, in this case (for both `OBE-Mean` and `OBE-Con`) we simply assign zero exploration bonus to all answers in the batch, thus recovering the regular `GRPO` objective.

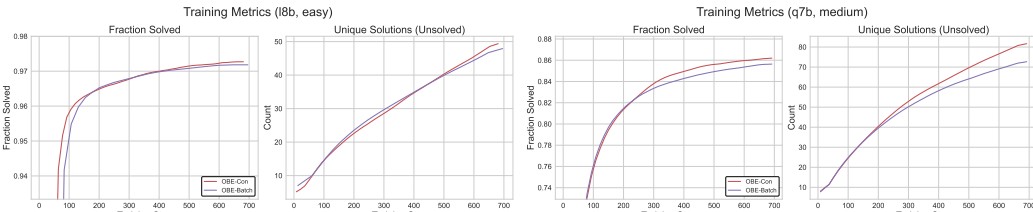

Figure 5: Training performance comparison between `OBE-Batch` and `OBE-Con`, with `Llama-3.1-8B-Instruct` on the easy dataset (left) and `Qwen-2.5-7B-Base` on the medium dataset (right). For each subplot: Left: fraction of questions solved so far; Right: number of different answers sampled on the questions that the model has yet to solve (i.e., sample one correct answer historically). The x-axis denotes the number of gradient updates as we train all models fully on-policy. We repeat each experiment with 3 different random seeds and plot the mean performance.

`OBE` **with a baseline generalizes towards test performance.** We compare these three variants with the `GRPO` baseline, and the results are summarized in Figs. 3, 4, 13 and 14. For the training performance, we observe that adding a baseline slightly hurts the training performance, but `OBE-Con` still outperforms `GRPO`. On the other hand, both `OBE-Mean` and `OBE-Con` consistently improve the test performance across different models and datasets. While `OBE-Mean` improves over UCB and `GRPO`, `OBE-Con` achieves the best frontier performance as it achieves the best pass@$k$ performance for all $k$'s in most of the settings. Another observation is that under our long training setup, vanilla RL (`GRPO`) sometimes suffers from overoptimization as the pass@$k$ degrades after a certain number of epochs for all $k$, while RL with exploration mitigates this issue. See Tables 3 and 4 for a quantitative comparison (we also compare with other exploration baseline such as entropy exploration (Cheng et al., 2025), and outcome-based exploration demonstrated superior performance).

However, in general, one should not expect global exploration to always achieve a high pass@$k$ when $k$ is large, especially at the end of the training. An expository example is that, to maximize the exploration bonus, the model can generate a batch of identical answers that currently has the least visitation counts. Indeed, in the theoretical RL literature, the goal of exploration is usually to return a policy that is deterministic (and optimal) (Azar et al., 2017) [1]. While in practice we observe that adding exploration bonus does provide better final pass@$k$ with large $k$ than vanilla RL, but the improvement can be small in certain cases.

### 3.3 BATCH EXPLORATION

The above issue suggests a fundamental but subtle difference between the goal of traditional RL exploration and the goal of exploration in the LLM reasoning setting. In traditional RL, the goal of exploration is to find the optimal policy which maximizes the expected return (corresponding to pass@1), while in the LLM reasoning setting, in addition to pass@1, sometimes we also care about the diversity of the generation. To encourage the model to generate diverse answers, we consider a different exploration strategy, *batch exploration*, which directly rewards the model to generate diverse answers regardless of their historical behavior. In particular, in batch exploration we propose the (`OBE-Batch`) objective, with $b(x, a_i)$ in Eq. (2) replaced by:

$$b_{\mathsf{batch}}(x, \{y_i, a_i\}_{i=1}^n)_i = -\frac{1}{n-1} \sum_{j \neq i} \mathbb{1}\{a_i = a_j\},$$

where we simply penalize each answer based on how repetitive it is in the batch. We remark that we also experimented with the positive version of the batch exploration bonus where we provide a bonus of 1 for unique answers in the batch, but our result shows that such positive batch exploration bonus does not provide meaningful improvement in either training or test results.

We summarize the experimental results in Figs. 5, 6, 15 and 16. We focus on comparing `OBE-Batch` with `OBE-Con` for a cleaner presentation since both methods outperform the `GRPO` baseline

---

[1]Note that this is not necessarily true with the KL-regularized objective. However, note that the optimal KL-regularized policy $\pi^\star \propto \pi_{\mathsf{base}} \cdot \exp\left(\frac{1}{\beta} r\right)$, and with a small $\beta$ the optimal policy can still be near deterministic.

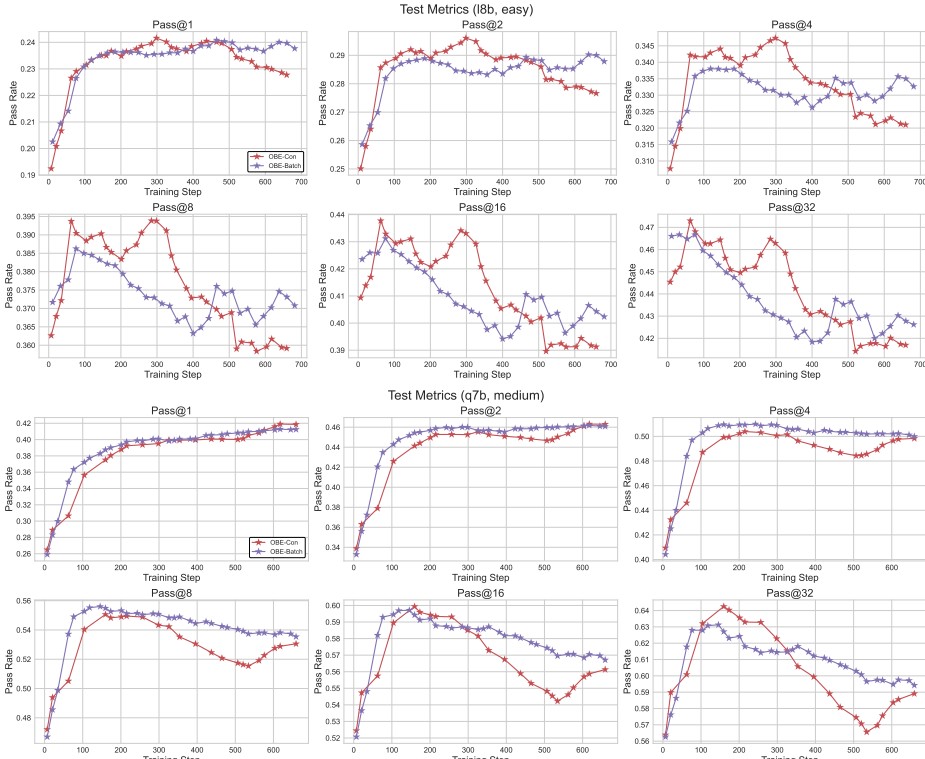

Figure 6: Test performance comparison between `OBE-Batch` and `OBE-Con`, with `Llama-3.1-8B-Instruct` on the easy dataset (top) and `Qwen-2.5-7B-Base` on the medium dataset (bottom). We report pass@$k$ for $k \in \{1, 2, 4, 8, 16, 32\}$ at every 20 training steps. We repeat each experiment with 3 different random seeds and plot the mean performance (see Section G for error bars). The metrics are calculated based on 32 samples per question during evaluation.

consistently. We observe that in general, `OBE-Batch` achieves worse performance during the training, as measured by both the fraction of questions solved and the number of different answers generated. However, note that the objective of `OBE-Batch` is not designed to explicitly optimize these two metrics. One can also consider an illustrative failure of batch exploration as the model keeps sampling the same $n$ distinct answers in each epoch for each question it could not solve yet (where $n$ is the batch size), and thus no real exploration is performed during training. As for test performance, in general `OBE-Batch` achieves similar peak pass@$k$ performance as `OBE-Con` (with slight degradation in some settings), but `OBE-Batch` consistently achieves better diversity at the end of the training, as measured by the pass@$k$ performance for large $k$. See Table 4 for a quantitative comparison. This suggests that batch exploration might be preferable if the objective is to achieve tradeoff between generation accuracy and diversity at test time.

## 4 ADDITIONAL ANALYSIS

### 4.1 HISTORICAL VS. BATCH EXPLORATION

In the previous section, we compared historical exploration and batch exploration in terms of their training dynamics. Overall, historical exploration is superior, as it solves more questions and accumulates more diverse answers over time. This is expected, since both metrics are inherently historical. In this section, we turn to the other aspects of exploration, focusing in particular on batch-level statistics. We defer the detailed results to Section D, and provide the takeaways from the results. We measure the token-level entropy and the number of distinct answers within a batch. We observe that `OBE-Batch` dominates on these metrics, but `OBE-Con` also outperforms Vanilla RL on these batch diversity metrics. Taken together with the ordering on historical training metrics (historical > batch > Vanilla RL), we remark that, despite their theoretical separation, historical and batch exploration are not mutually exclusive.

## 4.2 OUTCOME-BASED BANDITS

To better understand the role of exploration bonuses in our setting, we provide theoretical analysis in a simplified bandit model. While bandits are of course a coarse abstraction of LLM post-training, they have repeatedly yielded useful insights on sample efficiency, algorithmic tradeoffs, and problem difficulty in this context (Zhu et al., 2023; Rafailov et al., 2023; Azar et al., 2024; Chang et al., 2024; Song et al., 2024a). Here, we present a bandit formulation that captures the gap between the large reasoning-trace space and the much smaller answer space, and use it to explain why outcome-based UCB-style exploration is a principled strategy.

Concretely, we consider a stochastic bandit with a large set of arms $\mathcal{A}$, $|\mathcal{A}| = K$, but with an additional, much smaller set of outcomes $\mathcal{O}$, $|\mathcal{O}| = m \ll K$. Each arm $a \in \mathcal{A}$ maps to an outcome $\phi(a) \in \mathcal{O}$, and the reward of an arm depends only on its outcome: $\mathbb{E}[R \mid a] = \mu(\phi(a)) =: \mu(o)$. Intuitively, $\mathcal{A}$ corresponds to reasoning traces in the LLM setting, while $\mathcal{O}$ corresponds to final answers. This abstraction captures the fact that the answer space is far smaller than the trace space, where $K$ could be exponentially large. We consider the stochastic bandit setting instead of the contextual bandit setting for two reasons: first, the context space $\mathcal{X}$ in LLM reasoning is bounded, so even in the worst case without function approximation, we can treat it as $|\mathcal{X}|$ stochastic bandit problems. Second, our theory aims to focus on the relation between arms and outcomes, and thus the context is less relevant and is already well-studied in previous works.

A natural hope is that by exploiting the outcome-partition structure, we can achieve regret bounds that depend only on the number of outcomes $m$, not the number of arms $K$. This would mirror the LLM setting, where training on one reasoning trace should generalize to others that yield the same answer. However, without further assumptions, we show that this is not possible: even with outcome partitioning, the problem can remain as hard as the standard $K$-armed bandit.

**Theorem 4.1** (Lower bound, informal version of Theorem C.1). *For any algorithm, there exists an outcome-partitioned bandit instance with $K$ arms and $m$ outcomes such that the expected regret after $T$ rounds is at least $\Omega(\min\{T, K\})$.*

This lower bound highlights the need for an additional assumption: that policy updates on one arm can generalize to other arms yielding the same outcome. As we observe in Section 2.1, this assumption is naturally satisfied in the reasoning setting. Under such a generalization assumption, we recover the desired dependence on $m$, with an algorithm performing UCB-style exploration over the outcome space.

**Theorem 4.2** (Upper bound under generalization, informal version of Theorem C.3). *Under Assumption C.2, there exists an algorithm that achieves an expected regret after $T$ rounds of*

$$\mathbb{E}[R_T] \leq O\left(\sqrt{mT \log T}\right).$$

In other words, applying a UCB-style bonus at the outcome level recovers the standard regret bound of an $m$-armed bandit, as opposed to the original $K$-armed bandit. This provides theoretical justification for our outcome-based exploration algorithms.

## 5 CONCLUSION AND DISCUSSION

In this paper, we study the diversity degradation problem in LLM reasoning post-training through the analysis of RL as sampling. We observe two key phenomena: the transfer of diversity degradation and the tractability of outcome space in verifiable reasoning tasks. Based on these observations, we adopt the classical RL exploration strategy UCB in the outcome space, and a careful treatment between positive and negative exploration signals achieves improvement in test performance in the pass@$k$ metrics for all $k$. We also identify the distinction of the historical exploration in traditional RL and batch exploration that is more specific in the LLM reasoning setting, and derive the outcome-based batch exploration algorithm, which achieves better accuracy-diversity tradeoff at test time. Finally we provide further in-depth analysis on the connection of historical exploration and batch exploration, and a theoretical outcome-based bandit model that demonstrates the benefit of outcome-based exploration.

There are a few limitations of our work. First, our current algorithms only apply to the verifiable domain, and problems with a tractable outcome space, extending them to more general settings is an interesting future direction. Second, currently we only evaluate our methods on the single-turn benchmarks, and we believe exploration plays an even more significant role under the multi-turn settings.

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

# A    ADDITIONAL EXPERIMENTS

## A.1    BEYOND MATH DOMAINS

To verify that our method generalizes beyond math reasoning domain, we perform experiment on general reasoning dataset Webinstruct (Ma et al., 2025), which contains verifiable reasoning problems including domains such as physics, chemistry, biology, finance, etc. We first exclude the math questions from the dataset, and then subsample 4000 questions for training and another 400 questions for testing. We train the `Qwen-2.5-7B-Base` model and we report the test performance in Figure 7. We see both `OBE-Con` and `OBE-Batch` significantly outperforms `GRPO`, especially when $k$ is large. This result demonstrates the effectiveness and generalizability of our proposed methods.

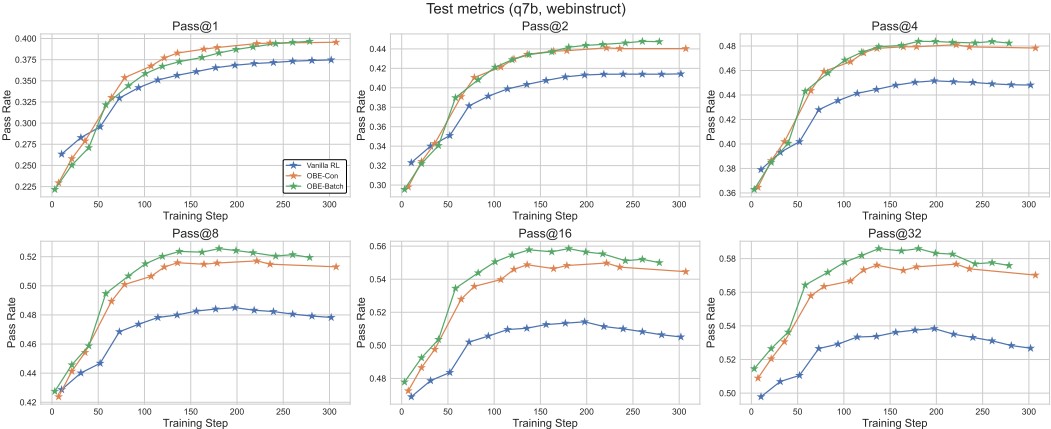

Figure 7: Test performance comparison between different `OBE` variants and the `GRPO` baseline, `Qwen-2.5-7B-Base` on the webinstruct dataset. We report pass@$k$ for $k \in \{1, 2, 4, 8, 16, 32\}$ at every 20 training steps. We repeat each experiment with 2 different random seeds and plot the mean performance. The metrics are calculated based on 32 samples per question during evaluation.

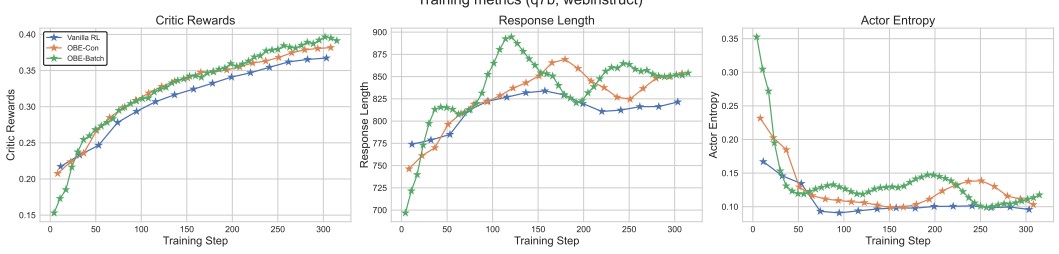

Figure 8: Training performance comparison between different `OBE` variants and the `GRPO` baseline, `Qwen-2.5-7B-Base` on the webinstruct dataset.

## A.2    ABLATIONS

In this section, we perform ablations over the hyperparameters introduced in `OBE-Con` and `OBE-Batch`.

### A.2.1    ABLATIONS ON THE BONUS COEFFICIENT

We track the test performance of `OBE-Con` and `OBE-Batch` with bonus coefficient $c \in \{0.01, 0.2, 1, 2\}$. We present the test results in Figs. 9 and 10. We observe similar patterns in both of the cases: first, a small coefficient such as $0.01$ brings very little improvement compared to `GRPO` because the objectives are almost identical when $c$ is small. A very large coefficient such as 2 causes policy collapse because the policy is encouraged to perform random exploration. A relatively large coefficient such as 1

degrades the performance in general but can improve the pass@$k$ when $k$ is large for `OBE-Batch`. Finally, the coefficient on the scale of our original choice of 0.2 bring the best overall performance with the best tradeoff between accuracy and exploration.

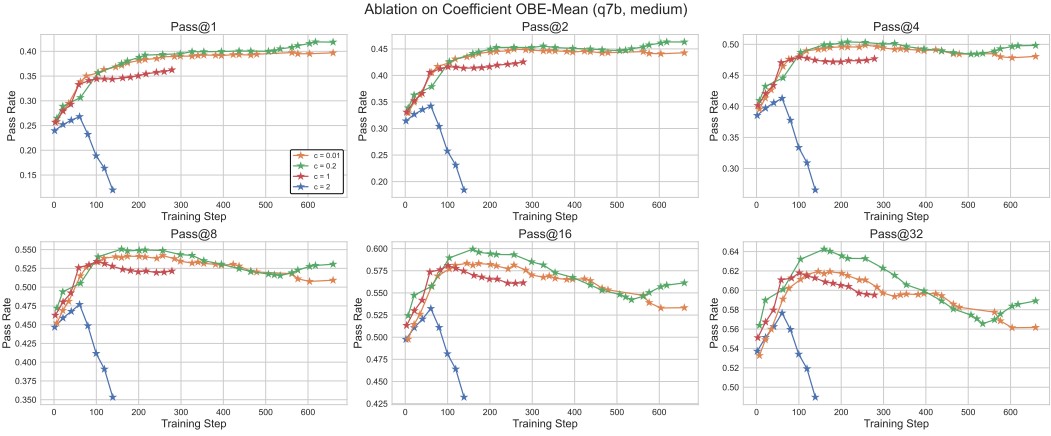

Figure 9: Test performance comparison between `OBE-Con` with different bonus coefficient $c \in \{0.01, 0.2, 1, 2\}$, `Qwen-2.5-7B-Base` on the medium dataset. We report pass@$k$ for $k \in \{1, 2, 4, 8, 16, 32\}$ at every 20 training steps. We repeat each experiment with 2 different random seeds and plot the mean performance. The metrics are calculated based on 32 samples per question during evaluation.

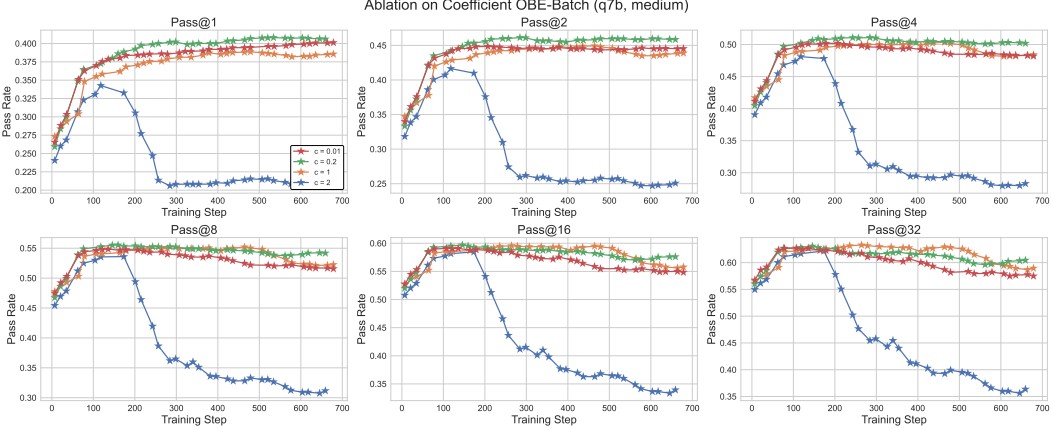

Figure 10: Test performance comparison between `OBE-Batch` with different bonus coefficient $c \in \{0.01, 0.2, 1, 2\}$, `Qwen-2.5-7B-Base` on the medium dataset. We report pass@$k$ for $k \in \{1, 2, 4, 8, 16, 32\}$ at every 20 training steps. We repeat each experiment with 2 different random seeds and plot the mean performance. The metrics are calculated based on 32 samples per question during evaluation.

### A.2.2 ABLATIONS ON THE CONSTANT BASELINE

We track the test performance of `OBE-Con` with constant baseline $b \in \{0.01, 0.1, 0.5, 1\}$. We present the test results in Figure 11. First note that a baseline that is large than 1 is not meaningful as the bonus is always bounded by 1. Overall the algorithm is less sensitive to the baseline than the coefficient, but a small baseline such as 0.01 almost recovers the performance of vanilla `OBE`. The performance increases as the baseline increases to 0.5, and 0.5 and 1 have very similar performance.

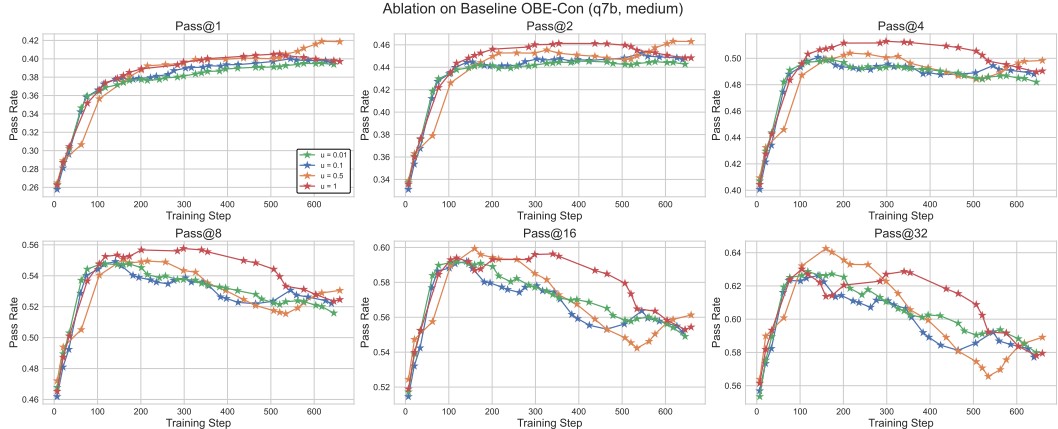

Figure 11: Test performance comparison between `OBE-Con` with different baselines $b \in \{0.01, 0.1, 0.5, 1\}$, `Qwen-2.5-7B-Base` on the medium dataset. We report pass@$k$ for $k \in \{1, 2, 4, 8, 16, 32\}$ at every 20 training steps. We repeat each experiment with 2 different random seeds and plot the mean performance. The metrics are calculated based on 32 samples per question during evaluation.

### A.3 TRAJECTORY SIMILARITY VS. OUTCOME SIMILARITY

We run an additional correlation analysis measuring whether outcome-level diversity tracks reasoning-level diversity on unsolved questions. For each model dataset pair, we pick 10 unsolved questions, and subsample $N = 100$ generations from these questions. For these samples, we estimated trajectory-level reasoning diversity using an LLM judge (GPT-4o-mini). Specifically, we perform pairwise comparisons to label whether two solutions follow the same or different reasoning strategies. This gave us a score $s_t \in [0, 1]$ for each pair. We then define the answer difference $s_a$, which is 1 if the pair has the same answer, and 0 if the pair has different answers.

We then computed the Spearman correlation between $s_t$ and $s_a$ for each of the 4 settings we have, where we obtain a score of 0.88 with a standard deviation of 0.03, indicating very strong correlation between the diversity of the outcome corresponds to the diversity of trajectory. Note that most mismatches between $s_t$ and $s_a$ are intra-problem, meaning that sometimes the same outcome can have slightly different trajectories. We detail the prompt below.

---

**LLM Judge Prompt**

**System Message**
You are a careful mathematical reasoning analyst. You are given a math problem and *two model-generated solutions*, A and B. Both solutions may be wrong. Your job is **not** to check correctness, but to decide whether A and B follow essentially the **same reasoning strategy** or **different reasoning strategies**.
Treat two solutions as the **same reasoning** if they:

- use the same main idea or decomposition of the problem,
- apply the same key equations or formulas in the same order,
- differ only by wording or superficial algebraic reordering.

Treat them as **different reasoning** if they:

- use different main approaches (e.g., geometry vs. algebra, casework vs. direct formula),
- decompose the problem into meaningfully different sub-problems,
- make different key substitutions or introduce different variables,
- pursue distinct algebraic chains that are not simple paraphrases.

Ignore differences in notation, variable names, or minor arithmetic slips.

---

> **User Message Template**
> Decide whether the two solutions use essentially the **same reasoning** or **different reasoning**. Then respond with a single JSON object:
>
> ```
> {
>   "label": "same" or "different",
>   "difference_score": float between 0 and 1,
>   "explanation": "1--3 sentence justification"
> }
> ```
> ——
> **Problem:**
> {QUESTION_TEXT}
> **Solution A:**
> {SOLUTION_A}
> **Solution B:**
> {SOLUTION_B}

## B  RELATED WORK

### B.1  DIVERSITY DEGRADATION IN LLM POST TRAINING

Reinforcement Learning has become the de facto method for finetuning large language models (LLMs) towards specific objectives, such as maximizing human preference (Ouyang et al., 2022), or improving the reasoning ability of LLMs (Jaech et al., 2024). In the reasoning domain, it has been shown that simply rewarding the model based on the correctness of the final answer, without any intermediate reward, can significantly improve the final accuracy (Shao et al., 2024; Guo et al., 2025). However, it has been observed that, during the RL training (or even SFT), the diversity of the generations decreases significantly (Song et al., 2024b; Dang et al., 2025; Yue et al., 2025; Wu et al., 2025), as measured with the pass@$k$ metric. In the non-reasoning domain, similar observations have also been made, where post-training improves the performance of the model on the main metric, but at the cost of losing diversity, measured by either semantic or syntactic metrics (Kirk et al., 2023; O'Mahony et al., 2024; Yun et al., 2025).

### B.2  EXPLORATION IN LLM POST TRAINING

Enhancing exploration during RL training has been considered the key towards addressing diversity issues during either training or testing. In the preference fine-tuning domain, Xie et al. (2024); Cen et al. (2024); Zhang et al. (2024a) propose to use the likelihood of the base model as an exploration bonus. Lanchantin et al. (2025) proposes to label ranking of the data based on their diversity in the preference learning process. Xiong et al. (2023); Bai et al. (2025) theoretically analyzes the guarantees of RL with exploration under the linear setting. Liu et al. (2025) promotes multi-turn reasoning and self-correction by prompting the model with "try again". Exploration has also been applied in the reasoning domain very recently. (Gao et al., 2025) directly uses Random Network Distillation (Burda et al., 2018), a canonical exploration bonus in Deep RL, as an exploration bonus to encourage the model to explore different traces, which, however, suffers from the growing length of the generations. Cheng et al. (2025); Zheng et al. (2025) proposes to leverage entropy to encourage exploration. Chen et al. (2025) leverages pass@$k$ training objective (Tang et al., 2025) to improve the batch diversity during training. Different from these contemporary works that mostly focus on the mechanical side of exploration, our method focuses on the semantic level of exploration, and thus is more interpretable. Meanwhile, our outcome-based method is complementary to all these methods, and can be potentially combined with them to further improve the diversity.

### B.3  EXPLORATION IN THEORETICAL RL

In the tabular setting, exploration has been studied extensively through count-based methods (Brafman & Tennenholtz, 2002; Kearns & Singh, 2002; Azar et al., 2017; Jin et al., 2018; Zhang et al., 2024b). These approaches rely on visitation counts to construct exploration bonuses. In linear MDPs, counts

are replaced by confidence sets in feature space (Yang & Wang, 2019; Jin et al., 2020; Ayoub et al., 2020; Zhou et al., 2021; Agarwal et al., 2023), with extensions to the discounted setting (Moulin et al., 2025) or model-based setting (Song & Sun, 2021) showing that the principle of optimism extends naturally from tabular counts to linear function approximation. The majority of these works share the same bonus-based exploration approach as our historical exploration method. Thompson sampling (Russo & Van Roy, 2014; 2018) is another popular exploration strategy that has been shown to achieve similar theoretical guarantees in both tabular and linear settings (Osband et al., 2013; 2016; Modi & Tewari, 2020). Finally, beyond tabular and linear settings, exploration in RL with general function approximation has been studied under various structural assumptions (Krishnamurthy et al., 2016; Jiang et al., 2017; Du et al., 2021; Foster et al., 2021). However, these methods do not enjoy computational efficiency as opposed to the bonus-based methods.

## C  THEORETICAL RESULTS

### C.1  PROBLEM SETUP: OUTCOME-BASED BANDIT

**Arms, outcomes, and partition.** Let $\mathcal{A}$ be a (large) set of arms with $|\mathcal{A}| = K$, and let $\mathcal{O} = \{1, \ldots, m\}$ be a (small) set of *outcomes* with $m \ll K$. There is an unknown partition mapping $\phi : \mathcal{A} \to \mathcal{O}$. For each outcome $o \in \mathcal{O}$, denote its class (preimage) and size by

$$\mathcal{A}_o := \phi^{-1}(o), \qquad s_o := |\mathcal{A}_o|, \qquad \sum_{o=1}^{m} s_o = K.$$

The partitions are mutually exclusive, i.e., for any $o, o' \in \mathcal{O}$ and $o \neq o'$ we have $\mathcal{A}_o \cap \mathcal{A}_{o'} = \emptyset$. Define its mass $p_o := s_o/K$. Partitions may be imbalanced (the $s_o$'s are arbitrary). We will also consider the balanced special case $s_o = K/m$ for all $o$.

**Reward.** Rewards depend only on the outcome: pulling arm $a \in \mathcal{A}$ yields a stochastic reward $R \in [0, 1]$ with

$$\mathbb{E}[R \mid a] = \mu\big(\phi(a)\big) =: \mu(o),$$

and we assume $R - \mu(o)$ is 1-sub-Gaussian (for example, Bernoulli). Let $\mu^\star := \max_{o \in \mathcal{O}} \mu(o)$ and let $o^\star \in \arg\max_o \mu(o)$ be an optimal outcome.

**Interaction protocol.** At each round $t = 1, 2, \ldots, T$, a (possibly randomized) policy $\pi$ selects an arm $A_t \in \mathcal{A}$ based on the history $\mathcal{H}_{t-1} := \{(A_s, O_s, R_s)\}_{s=1}^{t-1}$, where $O_s := \phi(A_s)$. The environment then reveals the outcome label $O_t = \phi(A_t)$ and draws the reward $R_t$ with mean $\mu(O_t)$. The filtration is the natural one generated by $\mathcal{H}_t$.

**Performance metric (pseudo-regret).** The pseudo-regret of policy $\pi$ over horizon $T$ is

$$R_T(\pi) := T\mu^\star - \mathbb{E}_\pi\left[\sum_{t=1}^{T} \mu\big(\phi(A_t)\big)\right] = \sum_{t=1}^{T} \mathbb{E}_\pi\left[\mu^\star - \mu(O_t)\right].$$

We will also use the discovery time of an outcome $o$,

$$\tau_o := \inf\{t \geq 1 : O_t = o\},$$

with the convention $\inf \emptyset = \infty$. These stopping times quantify the unavoidable delay before the learner first encounters a given outcome under the no-generalization constraint.

**LLM reasoning interpretation.** Each arm $a \in \mathcal{A}$ corresponds to a full reasoning trace; the mapping $\phi(a)$ is its final answer (outcome), and the reward is the verifiable correctness of that answer. Although the trace space is large, the outcome space $\mathcal{O}$ is small.

### C.2  LOWER BOUND

We first show that, even with the small outcome space, in the worst case any algorithm can not avoid paying a regret that is polynomial in the number of arms $K$, without any additional assumption.

**Theorem C.1** (Lower bound for outcome-based bandit). *Fix $K \geq 2$ and a partition $\{\mathcal{A}_o\}_{o=1}^m$ with class sizes $s_o = |\mathcal{A}_o|$ and a unique optimal outcome $o_\star$ of size $s_\star \in \{1, \dots, K\}$. Consider Bernoulli rewards with means $\mu(o_\star) = \frac{1}{2} + \Delta$ and $\mu(o) = \frac{1}{2}$ for all $o \neq o_\star$, where $\Delta \in (0, \frac{1}{2}]$. There exists a universal constant $c > 0$ such that for any (possibly randomized) algorithm $\widetilde{\pi}$ and any horizon $T$, there exists some fixed instance $I^\star$ such that*

$$\mathbb{E}\left[R_T(\widetilde{\pi}; I^\star)\right] \geq c\,\Delta \cdot \min\left\{T, \frac{K}{s_\star}\right\}.$$

**Proof.** Draw an instance by placing the $s_\star$ optimal arms uniformly at random among the $K$ arm indices (keeping all other classes fixed). By Lemma C.2, it suffices to lower-bound the expected regret of an *arbitrary deterministic* policy $\pi$ under this distribution; the resulting lower bound then holds for some fixed instance $I^\star$ against any randomized algorithm $\widetilde{\pi}$.

Let $\tau_\star$ be the index of the first pull from $\mathcal{A}_{o_\star}$. Before time $\tau_\star$ every reward is $\Delta$-suboptimal in expectation, hence

$$\mathbb{E}[R_T(\pi)] \geq \Delta \cdot \mathbb{E}\left[\min\{T, \tau_\star - 1\}\right] = \Delta \sum_{t=1}^{T} \Pr(\tau_\star > t).$$

Under the random placement, the event $\{\tau_\star > t\}$ is "no optimal arm among the first $t$ draws without replacement," so

$$\Pr(\tau_\star > t) = \frac{\frac{(K-s_\star)!}{(K-s_\star-t)!}}{\frac{K!}{(K-t)!}} = \prod_{i=0}^{t-1} \frac{K - s_\star - i}{K - i}.$$

For any $t \leq K/2$ we have

$$\frac{K - s_\star - i}{K - i} \geq \frac{K - s_\star - t}{K - t} \geq 1 - \frac{2s_\star}{K},$$

hence $\Pr(\tau_\star > t) \geq \left(1 - \frac{2s_\star}{K}\right)^t$. Set $t^\star := \min\left\{T, \lfloor \frac{K}{4s_\star} \rfloor\right\}$ (note $t^\star \leq K/2$ when $s_\star \leq K$). Then

$$\Pr(\tau_\star > t^\star) \geq \left(1 - \frac{2s_\star}{K}\right)^{K/(4s_\star)} \geq e^{-1/2},$$

and therefore

$$\mathbb{E}\left[\min\{T, \tau_\star - 1\}\right] = \sum_{t=1}^{T} \Pr(\tau_\star > t) \geq \sum_{t=1}^{t^\star} \Pr(\tau_\star > t) \geq t^\star \Pr(\tau_\star > t^\star) \geq c_0 \min\left\{T, \frac{K}{s_\star}\right\}$$

for a universal constant $c_0 > 0$ (e.g., $c_0 = e^{-1/2}/4$).

Now combining everything,

$$\mathbb{E}_I\left[R_T(\pi)\right] \geq \Delta \cdot \mathbb{E}\left[\min\{T, \tau_\star - 1\}\right] \geq c_0\,\Delta \cdot \min\left\{T, \frac{K}{s_\star}\right\}.$$

Then by Lemma C.2, for any randomized algorithm $\widetilde{\pi}$ there exists a fixed instance $I^\star$ in the support of the above distribution such that $\mathbb{E}\left[R_T(\widetilde{\pi}; I^\star)\right] \geq c\,\Delta \cdot \min\left\{T, \frac{K}{s_\star}\right\}$ with $c = c_0$. $\qquad\square$

## C.3 BALANCED PARTITIONS

Note that the previous lower bound comes from the imbalance of the partitions: the polynomial dependence on $K$ is due to the small (constant) size of the optimal class $s_\star$. We now show that if the partitions are balanced, i.e., $s_o = K/m$ for all $o \in \mathcal{O}$, then we can design an algorithm whose regret is independent of $K$.

**Assumption C.1** (Balanced partition). *Assume $|\mathcal{A}_o| = s = K/m$ for all $o \in \mathcal{O}$. Rewards depend only on the outcome and when an arm $a$ is pulled we observe its outcome $o = \phi(a)$ and the reward.*

---

**Algorithm 1** Balanced Outcome UCB

---

1: Initialize: $\mathcal{R}$ (set of discovered outcomes, initially $\emptyset$); $\mathrm{rep}[o]$ (representative arm for outcome $o$, initially undefined); $n_o \in \mathbb{N}$ and $\hat{\mu}_o \in \mathbb{R}$ for each discovered $o$ (both 0 initially); $U$ (set of unseen arms, initially $U = \mathcal{A}$).
2: **for** $t = 1, 2, \ldots, T$ **do**
3:     **if** $|\mathcal{R}| < m$ **then**                                                      ▷ Discovery phase
4:         Pick $A_t \sim \mathrm{Uniform}(U)$; $U \leftarrow U \setminus \{A_t\}$
5:         Pull $A_t$; observe outcome $o = \phi(A_t)$ and reward $r_t$
6:         **if** $o \notin \mathcal{R}$ **then**
7:             $\mathcal{R} \leftarrow \mathcal{R} \cup \{o\}$;   $\mathrm{rep}[o] \leftarrow A_t$
8:         $n_o \leftarrow n_o + 1$;   $\hat{\mu}_o \leftarrow \hat{\mu}_o + \dfrac{r_t - \hat{\mu}_o}{n_o}$
9:     **else**                                                         ▷ Outcome-level UCB
10:         For each $o \in \mathcal{R}$, set $\mathrm{UCB}_t(o) \leftarrow \hat{\mu}_o + \sqrt{\dfrac{2 \log t}{\max\{1, n_o\}}}$
11:         $o_t \in \arg\max_{o \in \mathcal{R}} \; \mathrm{UCB}_t(o)$
12:         Pull $\mathrm{rep}[o_t]$; observe $r_t$;    $n_{o_t} \leftarrow n_{o_t} + 1$;   $\hat{\mu}_{o_t} \leftarrow \hat{\mu}_{o_t} + \dfrac{r_t - \hat{\mu}_{o_t}}{n_{o_t}}$

---

**Theorem C.2** (Upper bound under Assumption C.1). *Assuming Assumption C.1, Algorithm 1 satisfies*

$$\mathbb{E}[R_T] \leq O(\sqrt{mT \log T}).$$

**Proof.** We decompose regret into a *discovery* part (before all $m$ outcomes have been observed) and a *bandit* part (afterwards, when we run UCB on the $m$ outcome representatives).

Let $\tau_{\mathrm{disc}}$ be the first round at which the set of observed outcomes equals $\mathcal{O}$ (i.e., the time to discover all $m$ outcomes). Let $o^\star$ be an optimal outcome with mean $\mu^\star$, and write $\Delta_o := \mu^\star - \mu(o) \in [0, 1]$.

While $t < \tau_{\mathrm{disc}}$ the algorithm draws *unseen* arms uniformly without replacement. Because the partition is balanced, the revealed outcome sequence is distributed as a uniformly random permutation of a multiset that contains exactly $K/m$ copies of each label. Let $T_{\mathrm{cc}}$ denote the coupon-collector time to see all $m$ labels *under i.i.d. sampling with replacement* where each label has probability $1/m$ per draw. We will show (Lemma C.3) that

$$\mathbb{E}[\tau_{\mathrm{disc}}] \leq \mathbb{E}[T_{\mathrm{cc}}] \leq m(\log m + 1).$$

Since per-round pseudo-regret is at most 1 (rewards lie in $[0, 1]$), the discovery contribution satisfies

$$\mathbb{E}\left[ \sum_{t=1}^{\min\{T, \tau_{\mathrm{disc}} - 1\}} (\mu^\star - \mu(O_t)) \right] \leq \mathbb{E}[\min\{T, \tau_{\mathrm{disc}} - 1\}] \leq \mathbb{E}[\tau_{\mathrm{disc}}] \leq m(\log m + 1). \quad (3)$$

Then at time $\tau_{\mathrm{disc}}$ the algorithm has stored one *representative* arm per outcome. From that time onward it never samples unseen arms; it only chooses among the $m$ representatives. Because rewards depend only on outcome, this reduces *exactly* to an $m$-armed stochastic bandit. Let $T' := \max\{0, T - \tau_{\mathrm{disc}} + 1\}$ be the number of post-discovery rounds (random, but $T' \leq T$). The algorithm runs UCB on the $m$ arms with index $\hat{\mu}_o + \sqrt{(2 \log t)/n_o}$ as specified in Algorithm 1. We will bound the regret in these $T'$ rounds by a standard UCB bound (Lemma C.4):

$$\mathbb{E}\left[ \sum_{t=\tau_{\mathrm{disc}}}^{T} (\mu^\star - \mu(O_t)) \;\Big|\; \tau_{\mathrm{disc}} \right] \leq O\left( \sqrt{m T' \log T} \right)$$

Taking expectations and using $T' \leq T$ yields

$$\mathbb{E}\left[ \sum_{t=\tau_{\mathrm{disc}}}^{T} (\mu^\star - \mu(O_t)) \right] \leq O\left( \sqrt{m T \log T} \right). \quad (4)$$

---

**Algorithm 2** Partition-Aware UCB under Strong Generalization

---

1: Initialize: $\mathcal{R}$ (set of discovered outcomes, initially $\emptyset$); rep[$o$] (representative arm for $o$, undefined until discovery); $n_o \in \mathbb{N}$, $\hat{\mu}_o \in \mathbb{R}$ for $o \in \mathcal{R}$ (both 0 initially); $U$ (pool of arms eligible for fresh probes, initially $U = \mathcal{A}$).
2: **for** $t = 1, 2, \ldots, T$ **do**
3:     **if** $|\mathcal{R}| < m$ **then**                                              ▷ Discovery phase
4:         Pick any $A_t \in U$; pull $A_t$; observe $(o = \phi(A_t), r_t)$
5:         **if** $o \notin \mathcal{R}$ **then**
6:             $\mathcal{R} \leftarrow \mathcal{R} \cup \{o\}$;   rep[$o$] $\leftarrow A_t$
7:             $n_o \leftarrow 1$;   $\hat{\mu}_o \leftarrow r_t$
8:             $U \leftarrow U \setminus \mathcal{A}_o$
9:         **else**
10:             $n_o \leftarrow n_o + 1$;   $\hat{\mu}_o \leftarrow \hat{\mu}_o + \dfrac{r_t - \hat{\mu}_o}{n_o}$
11:     **else**                                                   ▷ Outcome-level UCB
12:         For each $o \in \mathcal{R}$, set $\mathrm{UCB}_t(o) \leftarrow \hat{\mu}_o + \sqrt{\dfrac{2 \log t}{\max\{1, n_o\}}}$
13:         $o_t \in \arg\max_{o \in \mathcal{R}} \mathrm{UCB}_t(o)$
14:         Pull rep[$o_t$]; observe $r_t$; update $n_{o_t}, \hat{\mu}_{o_t}$

---

And we conclude the proof by summing Eq. (3) and Eq. (4), but noting that Eq. (4) is the dominant term. $\qquad\square$

## C.4    INBALANCED PARTITION UNDER STRONG GENERALIZATION

Our previous results are all established under a non-generalization scenario, where the update of arms in the same outcome partition is independent. This is not realistic under the LLM reasoning setting, since the update of one reasoning path (arm) can affect the other paths that lead to the same answer (outcome). We now show that this generalization capability is the key to demonstrating the benefit of outcome-based exploration, where previously under the worst case scenario the regret must depend on $K$. We start with a strong generalization assumption, where we assume that once an outcome is observed, the entire class of arms leading to that outcome is identified.

**Assumption C.2** (Strong generalization). *When the learner pulls an arm $a$ and observes the outcome $o = \phi(a)$, it henceforth knows the entire preimage $\mathcal{A}_o = \phi^{-1}(o)$ and can (i) route to $o$ by selecting any arm in $\mathcal{A}_o$, and (ii) exclude $\mathcal{A}_o$ from future fresh probes. For any outcome $o$ not yet observed, the learner cannot route to $\mathcal{A}_o$ nor exclude it a priori.*

Intuitively, with strong generalization, once an outcome is observed, the whole outcome partition is discovered, and then it can be treated as a single arm in the future. Then the problem reduces to an $m$-armed bandit problem. We provide the algorithm in Algorithm 2, which enjoys the following regret guarantee.

**Theorem C.3** (Regret upper bound under strong generalization). *Assuming Assumption C.2, Algorithm 2 satisfies:*

$$\mathbb{E}[R_T] \leq O\left(\sqrt{m\, T\, \log T}\right),$$

**Proof.** Let $\tau_{\mathrm{disc}}$ be the (random) time the set of discovered outcomes first equals $\mathcal{O}$. By construction, as soon as a new outcome $o$ is observed, the algorithm removes the entire class $\mathcal{A}_o$ from the fresh-probe pool $U$. Hence each subsequent fresh probe must land in an outcome not yet discovered. Therefore $\tau_{\mathrm{disc}} \leq m$ almost surely: at most one fresh probe per outcome is needed. Per-round pseudo-regret is at most 1, so the discovery contribution is bounded deterministically by

$$\sum_{t=1}^{\min\{T, \tau_{\mathrm{disc}} - 1\}} (\mu^\star - \mu(O_t)) \leq \min\{T, m\} \leq m.$$

At time $\tau_{\mathrm{disc}}$ the algorithm has one representative per outcome. From then on it only selects among these $m$ representatives using the UCB index $\hat{\mu}_o + \sqrt{(2 \log t)/n_o}$. This reduces exactly to an

---

---

**Algorithm 3** Soft-Exclusion Outcome-UCB

---

1: Initialize: $\mathcal{R}$ (discovered outcomes; initially $\emptyset$); $\text{rep}[o]$ (representative of $o$; undefined until discovery); $(n_o, \hat{\mu}_o)$ for $o \in \mathcal{R}$ (both 0 initially); $U$ (pool available for fresh probes; initially $U = \mathcal{A}$).
2: **for** $t = 1, 2, \ldots, T$ **do**
3:     **if** $|\mathcal{R}| < m$ and we are in *discovery mode* **then**
4:         Draw $A_t$ uniformly from $U$, pull it, observe $(o = \phi(A_t), r_t)$
5:         **if** $o \notin \mathcal{R}$ **then**
6:             $\mathcal{R} \leftarrow \mathcal{R} \cup \{o\}$;   $\text{rep}[o] \leftarrow A_t$;   $n_o \leftarrow 1$;   $\hat{\mu}_o \leftarrow r_t$
7:             (Soft exclusion) remove any $E_o \subseteq \mathcal{A}_o$ with $|E_o| = \rho_o s_o$
8:             $U \leftarrow U \setminus E_o$
9:         **else**
10:             $n_o \leftarrow n_o + 1$;   $\hat{\mu}_o \leftarrow \hat{\mu}_o + \dfrac{r_t - \hat{\mu}_o}{n_o}$
11:     **else**
12:         For each $o \in \mathcal{R}$, set $\text{UCB}_t(o) := \hat{\mu}_o + \sqrt{\dfrac{2 \log t}{\max\{1, n_o\}}}$
13:         $o_t \in \arg\max_{o \in \mathcal{R}} \ \text{UCB}_t(o)$
14:         pull $\text{rep}[o_t]$; observe $r_t$; update $n_{o_t}, \hat{\mu}_{o_t}$

---

$m$-armed stochastic bandit. By Lemma C.4, the post-discovery outcome-UCB satisfies

$$\mathbb{E}\Big[ \sum_{t=\tau_{\text{disc}}}^{T} (\mu^\star - \mu(O_t)) \Big] \leq O\Big( \sqrt{m T \log T} \Big).$$

Again by noting that the post-discovery contribution is the dominant term, we complete the proof. $\square$

### C.5   SOFT GENERALIZATION: ALGORITHM AND UPPER BOUND

The above example of strong generalization is idealized as it is unreasonable to believe that one witness of the outcome is sufficient for generalization towards all related arms. We now consider a more realistic *soft generalization* model. Intuitively, soft generalization states that instead of being able to exclude the entire class $\mathcal{A}_o$ upon first observing outcome $o$, the learner can only exclude a fraction of $\mathcal{A}_o$.

**Assumption C.3** (Soft generalization)**.** *Upon first observing an outcome $o$, the learner can (i) route perfectly to $o$ by re-pulling the observed arm; and (ii) exclude only a fraction of $\rho_o \in [0, 1]$ of the arms in $\mathcal{A}_o$ from future fresh probes.*

When $\rho_o = 1$ for all $o$, this reduces to strong generalization (Assumption C.2); when $\rho_o = 0$ for all $o$, this reduces to no generalization. Note that this assumption is rather unconventional: it is an assumption on the learner instead of the environment. However, this is a more suitable assumption here because it is indeed trying to model the generalization capability of the learner (LLM). With this we provide the soft-exclusion outcome-UCB algorithm in Algorithm 3 and analyze its performance.

**Theorem C.4** (Upper bound under soft generalization)**.** *Assuming soft generalization (Assumption C.3), Algorithm 3 satisfies*

$$\mathbb{E}[R_T] \leq O\left( \frac{1 - \bar{\rho}}{p_{o_\star}} + \sqrt{m T \log T} \right),$$

*where $\bar{\rho} := \sum_{o \neq o_\star} \rho_o p_o \in [0, 1)$.*

To prove this theorem, we first prove the stopping time that we discover the optimal outcome:

**Lemma C.1** (First hit of the optimal outcome)**.** *With Algorithm 3, let $\tau_\star$ be the index (number of fresh probes) of the first pull that lands in $\mathcal{A}_{o_\star}$. Then*

$$\mathbb{E}[\tau_\star] \leq \frac{1 - \bar{\rho}}{p_{o_\star}}.$$

**Proof.** Let $U_t$ denote the pool of arms available for fresh probes just *before* the $t$-th fresh probe, and let $E$ be the (random) union of all excluded sets $E_o$ over non-optimal outcomes $o \neq o_\star$ that have been discovered prior to time $\tau_\star$. By definition of soft exclusion, $|E| \leq \sum_{o \neq o_\star} \rho_o s_o = K\bar{\rho}$.

Fix any history up to time $t - 1$ with $\tau_\star > t - 1$. Then no arm from $\mathcal{A}_{o_\star}$ has been drawn yet, so all $s_{o_\star}$ optimal arms remain in $U_t$. The next fresh probe samples uniformly from $U_t$, hence

$$\Pr(\text{hit } \mathcal{A}_{o_\star} \text{ at probe } t \mid \tau_\star > t - 1, \ \mathcal{H}_{t-1}) = \frac{s_{o_\star}}{|U_t|} \geq \frac{s_{o_\star}}{K - |E|} \geq \frac{p_{o_\star}}{1 - \bar{\rho}}.$$

The first inequality uses $|U_t| \leq K - |E|$ (we can only remove arms, via exclusions and previous draws), and the second uses $|E| \leq K\bar{\rho}$. Therefore the conditional success probability at each probe is *uniformly* lower bounded by $p_{o_\star}/(1 - \bar{\rho})$, regardless of the discovery order of non-optimal outcomes.

Let $G$ be a geometric random variable with parameter $p_{o_\star}/(1 - \bar{\rho})$ (counting the number of trials until the first success). By the usual domination argument for inhomogeneous Bernoulli sequences with per-trial success probability bounded below,[2] we have $\tau_\star \preceq G$ (stochastic domination). Taking expectations gives $\mathbb{E}[\tau_\star] \leq (1 - \bar{\rho})/p_{o_\star}$. $\qquad\square$

**Proof of Theorem C.4.** Decompose the pseudo-regret as

$$R_T = \sum_{t=1}^{\min\{T, \tau_\star - 1\}} (\mu^\star - \mu(O_t)) + \sum_{t=\tau_\star}^{T} (\mu^\star - \mu(O_t)).$$

For the first term, we have

$$\mathbb{E}\Big[ \sum_{t=1}^{\min\{T, \tau_\star - 1\}} (\mu^\star - \mu(O_t)) \Big] \leq \mathbb{E}[\min\{T, \tau_\star - 1\}] \leq \mathbb{E}[\tau_\star] \leq \frac{1 - \bar{\rho}}{p_{o_\star}}$$

by Lemma C.1. For the second term, apply Lemma C.4. Summing the bounds gives the claim. $\quad\square$

**Remark C.1.** For all algorithms in this section, we separate the exploration phase (discovering outcomes) and the exploitation phase (outcome-level UCB). This is mainly for ease of presentation and analysis. In practice, the exploration phase can happen naturally with the outcome-level UCB, as the algorithm will always seek unpicked outcomes due to the design of the UCB term.

C.6 TECHNICAL LEMMAS

**Lemma C.2.** *Let $\Pi_{\mathrm{det}}$ be a (possibly infinite) set of deterministic algorithms (policies), let $\mathcal{I}$ be a set of instances, and let $L : \Pi_{\mathrm{det}} \times \mathcal{I} \to \mathbb{R}_{\geq 0}$ be any (measurable) loss functional. Write $\Delta(\Pi_{\mathrm{det}})$ and $\Delta(\mathcal{I})$ for distributions over algorithms and instances, respectively. Then*

$$\inf_{\widetilde{\pi} \in \Delta(\Pi_{\mathrm{det}})} \sup_{I \in \mathcal{I}} \mathbb{E}\big[ L(\widetilde{\pi}, I) \big] \ \geq \ \sup_{\mathcal{D} \in \Delta(\mathcal{I})} \inf_{\pi \in \Pi_{\mathrm{det}}} \mathbb{E}_{I \sim \mathcal{D}}\big[ L(\pi, I) \big].$$

**Lemma C.3** (Coupon-collector coupling (Motwani & Raghavan, 1996)). *In the balanced partition setting, let $\tau_{\mathrm{disc}}$ be the discovery time when drawing unseen arms uniformly without replacement. Let $T_{\mathrm{cc}}$ be the time to observe all $m$ labels under i.i.d. sampling with replacement with uniform label distribution $1/m$. Then $\mathbb{E}[\tau_{\mathrm{disc}}] \leq \mathbb{E}[T_{\mathrm{cc}}] = mH_m \leq m(\log m + 1)$.*

**Lemma C.4** (Auer et al. (2002)). *Consider an $m$-armed stochastic bandit with rewards in $[0, 1]$ that are 1-sub-Gaussian and means $\{\mu(o)\}_{o=1}^{m}$. Run UCB with index $\hat{\mu}_o + \sqrt{(2\log t)/n_o}$ for $t = 1, 2, \ldots, T$ (initializing each arm once). Then the pseudo-regret satisfies*

$$\mathbb{E}[R_T] \leq O\Big( \sqrt{mT \log T} \Big).$$

---

[2]Formally, if $X_t \in \{0, 1\}$ are conditionally independent given the past with $\Pr(X_t = 1 \mid \mathcal{H}_{t-1}) \geq \theta$ for all $t$, then $\sum_{u=1}^{t}(1 - X_u)$ is stochastically dominated by a sum of i.i.d. Bernoulli$(1 - \theta)$, and the first success time is dominated by $\mathrm{Geom}(\theta)$.

## D    ANALYSIS ON HISTORICAL VS. BATCH EXPLORATION

**Generation Entropy.**    Entropy has been used as a measure of model diversity and as a tool to encourage exploration (Cheng et al., 2025; Zheng et al., 2025). We compare token-level entropy averaged over the whole reasoning trajectory (including the outcomes), at training step 400 of the `Qwen-2.5-7B-Base` model on the medium dataset, trained with `GRPO`, `OBE-Con`, and `OBE-Batch`. For each method, we report the average entropy of correct and incorrect generations separately (Table 1). As expected, correct generations have lower entropy than incorrect ones. Among incorrect generations, however, `OBE-Batch` achieves substantially higher entropy than both `GRPO` and `OBE-Con`. Since entropy is measured on the current model rather than accumulated over training, this suggests that batch exploration yields generations with higher entropy, which reflects greater variability and potentially more diversity, when evaluated at a single checkpoint. That said, the absolute entropy values remain low across all methods, consistent with the fact that we do not explicitly optimize for entropy, unlike entropy-regularized exploration approaches.

Table 1: Entropy comparison of `GRPO`, `OBE-Con` and `OBE-Batch`, measured on correct generation, incorrect generation and all generations. We repeat for 2 random seeds and report the mean and standard deviation (in parentheses).

|           | Correct Generation | Incorrect Generation | All           |
| --------- | ------------------ | -------------------- | ------------- |
| GRPO      | 0.080 (0.01)       | 0.096 (0.04)         | 0.095 (0.02)  |
| OBE-Con   | 0.084 (0.01)       | 0.103 (0.03)         | 0.100 (0.02)  |
| OBE-Batch | 0.086 (0.01)       | 0.153 (0.07)         | 0.125 (0.03)  |

**Batch Generation Diversity.**    To directly measure batch-level diversity, we consider the number of distinct answers sampled within each batch. Results are shown in Table 2. As expected, `OBE-Batch` consistently produces more distinct answers than `OBE-Con`, since it directly optimizes for batch diversity.

Table 2: Comparison of different exploration strategies based on the number of different answers sampled in a batch with size of 8. We additionally cluster the statistics based on whether the question has been solved. We repeat for 2 random seeds and report the mean and standard deviation (in parentheses).

|           | Solved Question | Unsolved Question | All           |
| --------- | --------------- | ----------------- | ------------- |
| GRPO      | 2.279 (0.018)   | 4.805 (0.075)     | 2.883 (0.024) |
| OBE-Con   | 2.272 (0.020)   | 4.855 (0.084)     | 2.926 (0.035) |
| OBE-Batch | 2.284 (0.057)   | 5.390 (0.102)     | 3.230 (0.062) |

Finally, we remark on the interaction between historical and batch exploration. In principle, it is possible to construct counterexamples where historical exploration converges to a nearly deterministic policy—thus sacrificing test-time diversity—or where batch exploration cycles through a small set of answers without improving training dynamics. These pathologies highlight that the two notions are not guaranteed to substitute for one another. In practice, however, our empirical results suggest a complementary relationship: historical exploration, by encouraging broader coverage of the training space, naturally increases the diversity available to each batch, while batch exploration, by promoting variation within each batch, in turn helps prevent premature collapse during training. Taken together, these findings indicate that historical and batch exploration are not mutually exclusive.

# E    ADDITIONAL EXPERIMENT ON HARD DATASET

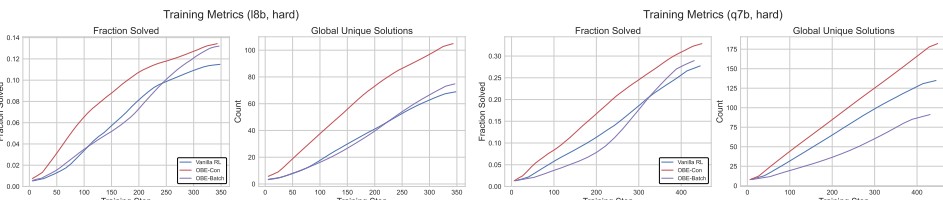

Figure 12: Training performance comparison between `OBE-Con`, `OBE-Batch` and the `GRPO` baseline, with `Llama-3.1-8B-Instruct` (left) and `Qwen-2.5-7B-Base` (right) on the hard dataset. For each subplot: Left: fraction of questions solved so far; Right: number of different answers sampled so far. The x-axis denotes the number of gradient updates as we train all models fully on-policy. We repeat each experiment with 3 different random seeds and plot the mean performance.

In this section, we compare the training performance between `OBE-Con`, `OBE-Batch` and the `GRPO` baseline on the hard dataset. For `Llama-3.1-8B-Instruct`, the dataset comprises 996 questions in total. For `Qwen-2.5-7B-Base`, the dataset comprises 753 questions in total. We observe the improvement of outcome-based exploration algorithm is even larger than the base model in this hard dataset, in both the number of questions solved and the number of unique solutions explored. However, we remark that this is more of a synthetic setting because training on this hard dataset does not bring meaningful improvement on the test dataset, for all of the baselines. Nevertheless, combining with the results from the medium dataset, our results indicate that exploration is more beneficial in the medium to hard training regime.

# F    OMITTED PLOTS

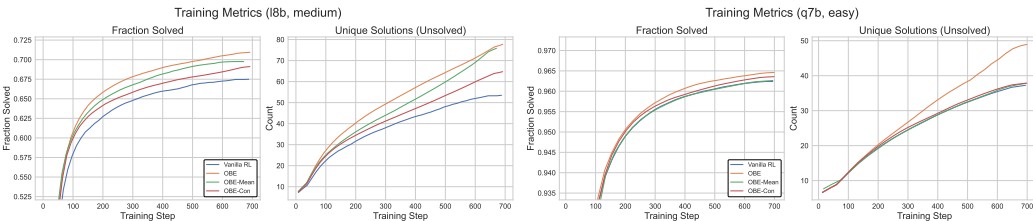

Figure 13: Training performance comparison between different `OBE` variants and the `GRPO` baseline, with `Llama-3.1-8B-Instruct` on the medium dataset (left) and `Qwen-2.5-7B-Base` on the easy dataset (right). For each subplot: Left: fraction of questions solved so far; Right: number of different answers sampled on the questions that the model has yet to solve (i.e., sample one correct answer historically). The x-axis denotes the number of gradient updates as we train all models fully on-policy. We repeat each experiment with 3 different random seeds and plot the mean performance.

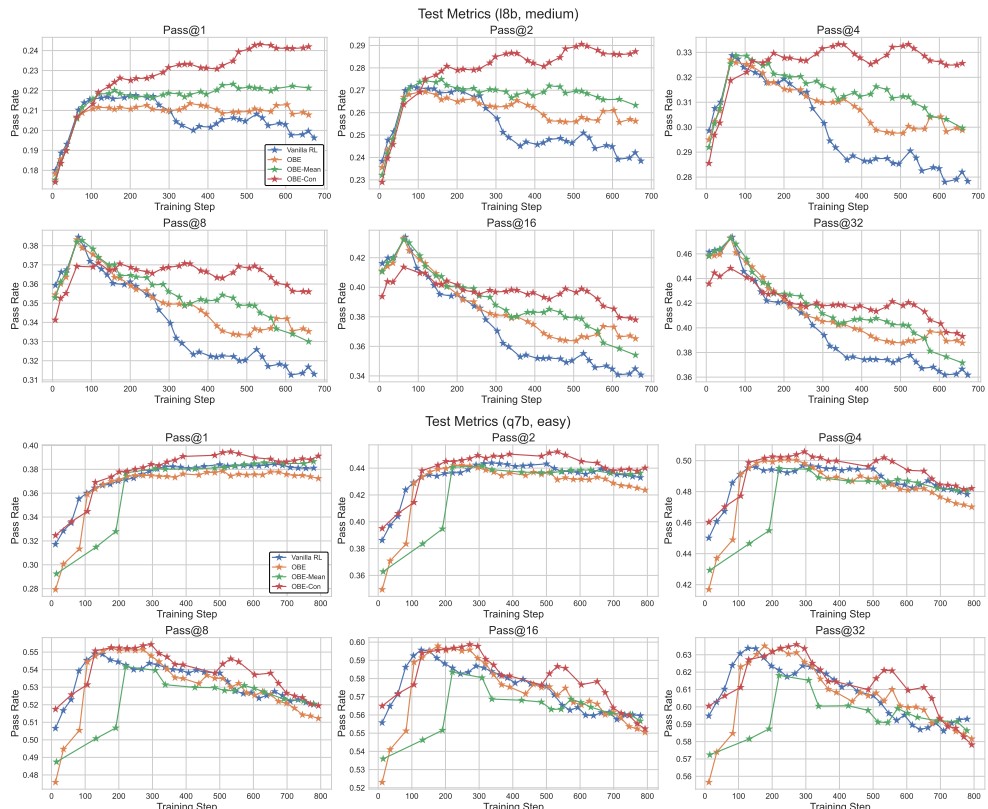

Figure 14: Test performance comparison between different `OBE` variants and the `GRPO` baseline, with `Llama-3.1-8B-Instruct` on the medium dataset (top) and `Qwen-2.5-7B-Base` on the easy dataset (bottom). We report pass@$k$ for $k \in \{1, 2, 4, 8, 16, 32\}$ at every 20 training steps. We repeat each experiment with 3 different random seeds and plot the mean performance (see Section G for error bars). The metrics are calculated based on 32 samples per question during evaluation.

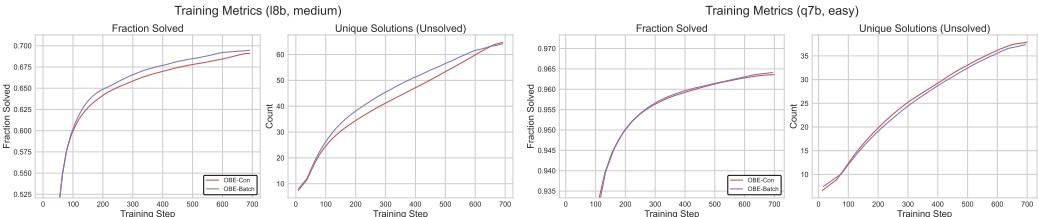

Figure 15: Training performance comparison between `OBE` and `OBE-Con`, `Llama-3.1-8B-Instruct` on the medium dataset (left) and `Qwen-2.5-7B-Base` on the easy dataset (right). For each subplot: Left: fraction of questions solved so far; Right: number of different answers sampled on the questions that the model has yet to solve (i.e., sample one correct answer historically). The x-axis denotes the number of gradient updates as we train all models fully on-policy. We repeat each experiment with 3 different random seeds and plot the mean performance.

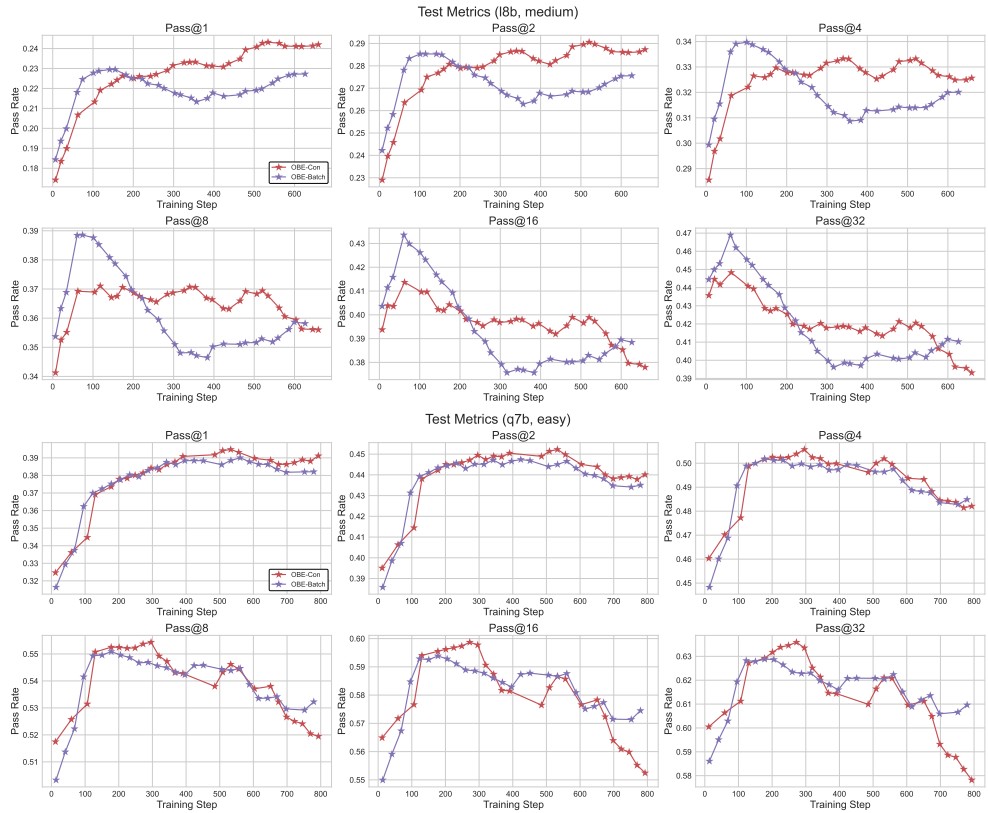

Figure 16: Test performance comparison between `OBE` and `OBE-Con`, with `Llama-3.1-8B-Instruct` on the medium dataset (top) and `Qwen-2.5-7B-Base` on the easy dataset (bottom). We report pass@$k$ for $k \in \{1, 2, 4, 8, 16, 32\}$ at every 20 training steps. We repeat each experiment with 3 different random seeds and plot the mean performance (see Section G for error bars). The metrics are calculated based on 32 samples per question during evaluation.

# G QUANTITATIVE RESULTS

In this section, we report the quantitative results from our experiments. We report the mean test performance and the one standard deviation. The pass@1 results are calculated based on the average of 32 generations. We compare with two baselines: Vanilla RL (`GRPO`) and entropy-based exploration (Entropy) (Cheng et al., 2025). We directly adopt the numbers reported in Cheng et al. (2025) since they also trained on the `DAPO` dataset, with `Qwen-2.5-7B-Base` model, and test on the same test sets (in Table 4). We report the raw statistics from Cheng et al. (2025), but we also report (in parentheses) the adjusted performance based on the difference in the reported Vanilla RL performances. Note that our method outperforms the baselines, but we remark that the goal of our paper is not to propose state-of-the-art methods, but instead to propose simple and flexible algorithmic interventions with theoretical support, to the diversity collapse problem that we identify through scientific experiments.

Table 3: Quantitative comparison of different baselines on pass@1 and pass@32 at the best checkpoint over three random seeds. We report mean and standard deviation in parentheses. The best mean results are in bold. Note that `OBE-Con` in general achieves the best peak performance.

| Method | Llama-3.1-8B-Instruct | | | | Qwen-2.5-7B-Base | | | |
| | Math | | DAPO | | Math | | DAPO | |
| | Pass@1 | Pass@32 | Pass@1 | Pass@32 | Pass@1 | Pass@32 | Pass@1 | Pass@32 |
|---|---|---|---|---|---|---|---|---|
| Vanilla RL | 0.231 (0.006) | 0.465 (0.031) | 0.218 (0.012) | **0.474** (0.014) | 0.385 (0.003) | 0.634 (0.008) | 0.403 (0.002) | 0.627 (0.004) |
| OBE | 0.236 (0.000) | 0.472 (0.024) | 0.214 (0.004) | 0.473 (0.014) | 0.379 (0.002) | 0.635 (0.012) | 0.397 (0.001) | 0.627 (0.005) |
| OBE-Mean | 0.239 (0.003) | 0.462 (0.003) | 0.223 (0.006) | 0.473 (0.011) | 0.387 (0.002) | 0.618 (0.001) | 0.407 (0.000) | 0.633 (0.008) |
| OBE-Con | **0.242** (0.003) | **0.473** (0.003) | **0.243** (0.005) | 0.448 (0.003) | **0.395** (0.001) | **0.636** (0.006) | **0.419** (0.001) | **0.642** (0.002) |
| OBE-Batch | 0.241 (0.001) | 0.467 (0.025) | 0.229 (0.003) | 0.469 (0.014) | 0.390 (0.007) | 0.629 (0.011) | 0.413 (0.008) | 0.631 (0.005) |

Table 4: Quantitative comparison of different baselines on pass@1 and pass@32 at the final checkpoint over three random seeds. We report mean and standard deviation in parentheses. The best mean results are in bold. Note that `OBE-Batch` in general achieves the best final performance in terms of pass@32. We also report the raw performance and adjusted performance from Cheng et al. (2025), denoted as Entropy.

| Method | Llama-3.1-8B-Instruct | | | | Qwen-2.5-7B-Base | | | |
| | Math | | DAPO | | Math | | DAPO | |
| | Pass@1 | Pass@32 | Pass@1 | Pass@32 | Pass@1 | Pass@32 | Pass@1 | Pass@32 |
|---|---|---|---|---|---|---|---|---|
| Vanilla RL | 0.215 (0.018) | 0.395 (0.017) | 0.196 (0.012) | 0.362 (0.034) | 0.381 (0.012) | 0.593 (0.006) | 0.399 (0.010) | 0.580 (0.013) |
| Entropy | - | - | - | - | - | - | 0.352 (0.414) | 0.575 (0.596) |
| OBE | 0.233 (0.003) | 0.425 (0.006) | 0.208 (0.008) | 0.388 (0.006) | 0.372 (0.013) | 0.582 (0.008) | 0.392 (0.007) | 0.580 (0.008) |
| OBE-Mean | 0.233 (0.003) | 0.414 (0.004) | 0.221 (0.008) | 0.372 (0.012) | 0.387 (0.002) | 0.586 (0.005) | 0.407 (0.006) | **0.603** (0.007) |
| OBE-Con | 0.228 (0.003) | 0.417 (0.007) | **0.242** (0.009) | 0.393 (0.005) | 0.391 (0.003) | 0.578 (0.007) | **0.419** (0.006) | 0.589 (0.006) |
| OBE-Batch | **0.238** (0.009) | **0.426** (0.011) | 0.227 (0.004) | **0.410** (0.009) | 0.382 (0.001) | **0.610** (0.001) | 0.412 (0.008) | 0.594 (0.010) |

## H    IMPLEMENTATION DETAILS

Our codebase is developed based on the `verl` codebase (Sheng et al., 2024). Thus we use the verl naming convention for the hyperparameters. For all our experiments, we use the hyperparameters in Table 5 unless otherwise specified. For all `Llama-3.1-8B-Instruct` experiments, we set bonus coefficient $c = 0.1$, and for all `Qwen-2.5-7B-Base` experiments, we set $c = 0.2$. For `OBE-Con`, we set $b_0 = 1$ for easy dataset and $b_0 = 0.5$ for medium dataset.

Table 5: Default hyperparameters for all baselines.

| Name | Value |
|------|-------|
| train batch size | 256 |
| learning rate | 1e-6 |
| ppo mini batch size | 256 |
| kl loss coef | 0.001 |
| entropy coeff | 0 |
| rollout.n | 8 |
| rollout.val_kwargs.temperature | 1 |

## I    LLM USAGE

We use LLMs to improve the writing of the paper (e.g., we feed certain paragraphs from the earlier draft and ask the LLMs to provide suggestions on writings). We also use LLMs to check typos or potential mathematical errors in our theoretical results. Finally, our codebase is developed with LLM coding assistants.

