# OpenReview forum: "Outcome-based Exploration for LLM Reasoning"
_ICLR.cc/2026/Conference — Submitted to ICLR 2026_

### Official Review · Reviewer_UQnj · 2025-10-24

**Soundness:** 3
**Presentation:** 3
**Contribution:** 3
**Rating:** 8
**Confidence:** 4

**Summary:**

The paper tackles diversity collapse during post-training RL for reasoning models and proposes shifting exploration to the outcome (answer) space rather than token trajectories. It introduces two complementary algorithms: historical exploration, which encourages infrequent answers via UCB-style bonuses, and batch exploration, which penalizes within-batch repetition of answers to promote test-time diversity. Experimental results demonstrate that their method enhances the reasoning without sacrificing the diversity during test time.

**Strengths:**

1. The authors thoroughly analyze diversity dynamics during training and identify a key causal issue, "the transfer of diversity degradation", where solving certain problems negatively affects the diversity of solutions generated for unsolved problems.

2. The authors introduce an additional reward term to enhance the diversity via UCB, and further introduce normalization combined with UCB to improve the test time performance. Finally, batch normalization of UCB to enhance the diversity during test time.

3. Theoretical and empirical results demonstrate the effectiveness of their methods.

**Weaknesses:**

1. There is a line of work that aims to mitigate diversity degradation using entropy-based methods [1] and pass@k-based training [2]. It would strengthen the paper if the authors conducted a more thorough comparison with these approaches—for example, by evaluating accuracy across different values of k, the number of unique solutions, and entropy.

2. Although the authors provide detailed accuracy results (I appreciate that), there is still a lack of analysis on the dynamics of reward, completion length, and entropy throughout training. It is also important to show how the UCB term influences the training process, perhaps by illustrating one or two specific cases on individual questions.

3. The authors fix c = 0.2, and use $b_0 = 1$ for the easy dataset and $b_0 = 0.5$ for the medium dataset. Yet, no ablation study is provided to justify these choices. As a result, the effect of these terms on training behavior, test performance, and solution diversity is still unclear.




[1] Wang, Shenzhi, et al. "Beyond the 80/20 rule: High-entropy minority tokens drive effective reinforcement learning for llm reasoning." arXiv preprint arXiv:2506.01939 (2025).

[2] Chen, Zhipeng, et al. "Pass@ k training for adaptively balancing exploration and exploitation of large reasoning models." arXiv preprint arXiv:2508.10751 (2025).

**Questions:**

1. On page 5, you state that “N(x,a) denotes the number of times answer a has been sampled for question x.” If a is a newly generated answer for x, is N(x,a) initialized as 0 or 1? If it is 0, the expression 1/N(x,a) becomes problematic, and it may be necessary to introduce a small \epsilon term for stability.

---

> ### Author Response · Authors · 2025-11-21
> **Author Rebuttal (1/2)**
>
> We are grateful for the reviewer’s insights. We outline our response to the comment below and we look forward to a constructive discussion period.
>
> > There is a line of work that aims to mitigate diversity degradation using entropy-based methods [1] and pass@k-based training [2]. It would strengthen the paper if the authors conducted a more thorough comparison with these approaches—for example, by evaluating accuracy across different values of k, the number of unique solutions, and entropy.
> >
>
> [1] Wang, Shenzhi, et al. "Beyond the 80/20 rule: High-entropy minority tokens drive effective reinforcement learning for llm reasoning." arXiv preprint arXiv:2506.01939 (2025).
>
> [2] Chen, Zhipeng, et al. "Pass@ k training for adaptively balancing exploration and exploitation of large reasoning models." arXiv preprint arXiv:2508.10751 (2025).
>
> We indeed compare with an entropy-based exploration method [3]. We record the result in Table 4, and we observe that our proposed method OBE-Mean and OBE-Batch outperforms [3]. For the rebuttal, we try to perform pass@k training with the native verl implementation of pass@k objective, but unfortunately, pass@k training collapses in our model/data settings. We will update if we obtain any meaningful results. We will also make sure to add the discussion of these two related papers.
>
> [3] Daixuan Cheng, Shaohan Huang, Xuekai Zhu, Bo Dai, Wayne Xin Zhao, Zhenliang Zhang, and Furu Wei. Reasoning with exploration: An entropy perspective. arXiv preprint arXiv:2506.14758, 2025.
>
> > Although the authors provide detailed accuracy results (I appreciate that), there is still a lack of analysis on the dynamics of reward, completion length, and entropy throughout training. It is also important to show how the UCB term influences the training process, perhaps by illustrating one or two specific cases on individual questions.
> >
>
> We appreciate the suggestions. In Appendix D, we provide comparisons of the final entropy and the number of different generations during training. For the rebuttal, we provide a comparison of training reward, generation length, and entropy throughout training in Fig. 8, Appendix A.1, in the revised submission, for our new experiment setting in webinstruct, a general reasoning dataset. We observe that the training reward of OBE variants increases more slowly at the beginning but quickly dominates GRPO baseline, and in general have longer generation length and entropy, but also have higher fluctuations during training.

---

> > ### Author Response · Authors · 2025-11-21
> > **Author Rebuttal (2/2)**
> >
> > > No ablation study is provided to justify our choice of hyperparameters.
> > >
> >
> > Thank you for bringing this up. We have conducted a small-scale hyperparameter search to ensure that our current hyperparameters are in the optimal range for performance. In the revised submission, we perform ablation over the bonus coefficient for OBE-Con and OBE-Batch, and ablation over the baseline value for OBE-Con. We provide a highlight of the results, and we refer the reviewer to Appendix A.2 for detailed experiment results.
> >
> > We track the test performance of OBE-Con and OBE-Batch with bonus coefficient $c \in \{0.01, 0.2, 1,2\}$. We observe similar patterns in both of the cases: first, a small coefficient ($c=0.01$) brings very little improvement compared to GRPO because the objectives are almost identical when $c$ is small. A very large coefficient ($c=2$) causes policy collapse because the policy is encouraged to perform random exploration. A relatively large coefficient ($c=1$) degrades the performance in general but can improve the pass@k when k is large for OBE-Batch. Finally, the coefficient on the scale of our original choice of 0.2 brings the best overall performance with the best tradeoff between accuracy and exploration.
> >
> > For the baseline, We track the test performance of OBE-Con with constant baseline $b \in \{0.01, 0.1, 0.5,1\}$. First note that a baseline that is large than 1 is not meaningful as the bonus is always bounded by 1. Overall the algorithm is less sensitive to the baseline than the coefficient, but a small baseline such as 0.01 almost recovers the performance of vanilla OBE. The performance increases as the baseline increases to 0.5, and 0.5 and 1 have very similar performance.
> >
> > > On page 5, you state that “N(x,a) denotes the number of times answer a has been sampled for question x.” If a is a newly generated answer for x, is N(x,a) initialized as 0 or 1? If it is 0, the expression 1/N(x,a) becomes problematic, and it may be necessary to introduce a small \epsilon term for stability.
> > >
> >
> > We initialize N(x,a) to be 0, so for newly generated answer $\sqrt{1/N(x,a)}$ is indeed unbounded. However, for the bonus, we define the bonus as  $\min \{1,\sqrt{1/N(x,a)}\}$  (see the expression below Eq. 2), so in any case, the bonus is always bounded by 1.

---

> > > ### Comment · Reviewer_UQnj · 2025-11-25
> > >
> > > Thank you for your response. I have no further questions.

---

> > > > ### Author Response · Authors · 2025-11-26
> > > >
> > > > We are glad to address the reviewer's questions and we appreciate the timely response and support from the reviewer.

---

### Official Review · Reviewer_bqjP · 2025-10-27

**Soundness:** 2
**Presentation:** 2
**Contribution:** 2
**Rating:** 2
**Confidence:** 3

**Summary:**

This paper introduces the UCB  reward mechanism—commonly used in traditional reinforcement learning to encourage exploration—into the reinforcement learning training process of LLMs. It proposes an exploration strategy based on the final answer, aiming to encourage the model to generate more diverse outputs.

**Strengths:**

- This method partially mitigates the trend of performance degradation on the test set as training progresses, exhibiting a certain regularization effect.

**Weaknesses:**

**From a methodological perspective:**

- If a model is already capable of correctly answering a given question, is it still meaningful to pursue answer-level diversity for that question? In most tasks (e.g., mathematical reasoning), the correct answer is typically unique. In such cases, "increasing answer diversity" may effectively equate to encouraging the generation of more diverse **incorrect answers**. This raises concerns about the validity of the optimization objective itself, which may even lead the model to waste exploration resources on low-quality or clearly incorrect outputs, thereby affecting training efficiency and final performance.

**From an experimental perspective:**

- The method introduces several hyperparameters (such as $c$ and $b_0$), yet the paper does not provide a systematic hyperparameter sensitivity analysis or ablation study.
- Experimental results show that, regardless of whether OBE is used, test performance tends to first increase and then decline, a phenomenon especially evident on simpler datasets. This trend resembles **overfitting** in supervised learning, suggesting that the underlying issue may stem from repeated training on limited data, leading to model memorization. The role of OBE seems to merely delay this process, without yielding significant improvements in peak performance compared to baseline methods (e.g., GRPO).

**Questions:**

As mentioned in the Weaknesses

---

> ### Author Response · Authors · 2025-11-21
> **Author Rebuttal (1/2)**
>
> Thank you for your thoughtful review. We provide our response to the reviewer’s comment below and we look forward to a constructive discussion period.
>
> > If a model is already capable of correctly answering a given question, is it still meaningful to pursue answer-level diversity for that question? In most tasks (e.g., mathematical reasoning), the correct answer is typically unique. In such cases, "increasing answer diversity" may effectively equate to encouraging the generation of more diverse incorrect answers. This raises concerns about the validity of the optimization objective itself, which may even lead the model to waste exploration resources on low-quality or clearly incorrect outputs, thereby affecting training efficiency and final performance.
> >
>
> The reason that we apply OBE exploration bonus even on the incorrect generation is to reduce the effect of undesirable “transfer of diversity degradation across questions” (section 2.1), i.e., to keep the diversity of the model even on solved questions so that it does not sharpen on certain wrong answers for unsolved questions.
>
> On the other hand, we agree with the reviewer that naively applying OBE might waste exploration resources on low-quality or clearly incorrect outputs, and thus we propose OBE-Mean/Con to tradeoff the positive and negative exploration signals during training (so we penalize the policy if it over-explores wrong answers), where OBE-Con consistently improves performance on all pass@k. Our empirical results also demonstrate the effectiveness of our proposed algorithms (OBE-Mean/Con). We refer the reviewer to the discussion on section 3.1.1 (now highlighted in green) for more details.
>
> > The method introduces several hyperparameters, yet the paper does not provide a systematic hyperparameter sensitivity analysis or ablation study.
> >
>
> Thank you for bringing this up. We have conducted a small-scale hyperparameter search to ensure that our current hyperparameters are in the optimal range for performance. In the revised submission, we perform ablation over the bonus coefficient for OBE-Con and OBE-Batch, and ablation over the baseline value for OBE-Con. We provide a highlight of the results, and we refer the reviewer to Appendix A.2 for detailed experiment results.
>
> We track the test performance of OBE-Con and OBE-Batch with bonus coefficient $c \in \{0.01, 0.2, 1,2\}$. We observe similar patterns in both of the cases: first, a small coefficient ($c=0.01$) brings very little improvement compared to GRPO because the objectives are almost identical when $c$ is small. A very large coefficient ($c=2$) causes policy collapse because the policy is encouraged to perform random exploration. A relatively large coefficient ($c=1$) degrades the performance in general but can improve the pass@k when k is large for OBE-Batch. Finally, the coefficient on the scale of our original choice of 0.2 brings the best overall performance with the best tradeoff between accuracy and exploration.
>
> For the baseline, We track the test performance of OBE-Con with constant baseline $b \in \{0.01, 0.1, 0.5,1\}$. First note that a baseline that is large than 1 is not meaningful as the bonus is always bounded by 1. Overall the algorithm is less sensitive to the baseline than the coefficient, but a small baseline such as 0.01 almost recovers the performance of vanilla OBE. The performance increases as the baseline increases to 0.5, and 0.5 and 1 have very similar performance.

---

> > ### Author Response · Authors · 2025-11-21
> > **Author Rebuttal (2/2)**
> >
> > > Experimental results show that, regardless of whether OBE is used, test performance tends to first increase and then decline, a phenomenon especially evident on simpler datasets. This trend resembles overfitting in supervised learning, suggesting that the underlying issue may stem from repeated training on limited data, leading to model memorization. The role of OBE seems to merely delay this process, without yielding significant improvements in peak performance compared to baseline methods (e.g., GRPO).
> > >
> >
> > We first clarify that, the most important performance metric, test accuracy (i.e., pass@1), consistently increases throughout the training procedure. We highlight that in the case of LLaMa model, pass@1 indeed goes down for GRPO, but with OBE variants pass@1 keeps increasing, demonstrating the importance of the exploration bonus.
> >
> >
> >
> > That said, we agree the other test metrics, pass@k when $k \neq 1$, tend to first increase then decrease during the training. However, this is expected, as previous work has shown that, to increase the performance of accuracy (i.e., pass@1), one has to tradeoff with diversity (e.g., pass@k) [1], and thus eventually pass@k ( $k \neq 1$) has to go down to obtain a better accuracy. In practice, one can early stop the training if the goal is to obtain a model with the best pass@k ( $k \neq 1$) rate. Finally, we remark that, the goal of OBE is not to obtain perpetual increase in pass@k over a large number of epochs (which is not the normal training procedure in practice nevertheless), but instead is to obtain a model that is better than vanilla RL training in both accuracy and diversity, during anytime of the training (except the very beginning of the training). We will make sure to emphasize this point in the final version of the paper.
> >
> > [1] Cui, Ganqu, et al. "The entropy mechanism of reinforcement learning for reasoning language models." *arXiv preprint arXiv:2505.22617* (2025).

---

> > ### Comment · Reviewer_bqjP · 2025-11-24
> >
> > - In practical applications, users are more concerned about achieving a sufficiently high pass@1 performance. Maintaining pass@k might not be as meaningful.
> > - Additionally, the experiments were conducted by training on a small dataset over multiple epochs, which might inherently favor your method. With a larger dataset and an appropriate curriculum learning setup, GRPO could also effectively scale up. [1]
> > - You still have not clearly explained the fundamental difference between encouraging answer diversity and encouraging the generation of incorrect answers.
> >
> > [1] Xu Y, Dong H, Wang L, et al. Scalable chain of thoughts via elastic reasoning[J]. arXiv preprint arXiv:2505.05315, 2025.

---

> > > ### Author Response · Authors · 2025-11-26
> > > **Response (1/2)**
> > >
> > > We thank the reviewer for their timely reply. Here is our response:
> > >
> > > > In practical applications, users are more concerned about achieving a sufficiently high pass@1 performance. Maintaining pass@k might not be as meaningful.
> > >
> > > We respectfully disagree with the statement. Achieving high pass@k is crucial in real-world production. Modern code assistants and evaluation harnesses are explicitly designed around multi-sample generation and validation; in these systems, pass@k is crucial, not only the probability that the first candidate happens to be correct. [OpenAI Codex, 2]
> > >
> > > In academia, improving pass@k is one of the most popular topic due to its connection to expanding reasoning capability beyond base models [3,4,5] and test-time performance [6,7]. Our work is one of the first paper that proposes an algorithmic intervention towards solving this important and challenging problem.
> > >
> > > That said, as the results demonstrate, our proposed method improves pass@1 in addition to pass@k, so our proposed also improves on the metric that the reviewer deems the most important.
> > >
> > > > Additionally, the experiments were conducted by training on a small dataset over multiple epochs, which might inherently favor your method. With a larger dataset and an appropriate curriculum learning setup, GRPO could also effectively scale up. [1]
> > >
> > > We clarify that we train on the standard datasets (MATH and DAPO) so our experiment is beyond small-scale dataset. We subsample the DAPO dataset (but still on the same scale) due to computational considerations but we believe the results hold the same if we train on the full dataset. Regarding the second half of the comment, our paper does not claim that GRPO does not effectively scale up, but adding exploration can further improve the performance in addition to GRPO’s gains. In addition, our method is orthogonal to the curriculum, so one can apply OBE and the curriculum at the same time.
> > >
> > > Training on multiple epochs also has an important implication towards future research as eventually we will be in a data-constrained setting where the amount of compute will surpass the amount of data by orders of magnitude. One of our key results is that applying exploration can mitigate overoptimization, which is essential in the data-constrained setting.

---

> > > > ### Author Response · Authors · 2025-11-26
> > > > **Response (2/2)**
> > > >
> > > > > You still have not clearly explained the fundamental difference between encouraging answer diversity and encouraging the generation of incorrect answers.
> > > >
> > > > We agree that encouraging answer diversity does encourage the model to generate incorrect answers (as there is only one correct answer). However, we again remind the reviewer that this is by design as it mitigates the “transfer of diversity degradation” problem (one of the key observation of our paper). Our results demonstrate that the strategy of encouraging answer diversity is very effective as it improves pass@k for all k, evidenced over multiple domains, models and datasets. The key, as our ablation experiments show, is to balance the correctness signal and the diversity signal.
> > > >
> > > > To further demonstrate the benefits of diversity, consider the following simplified example: suppose we have 10 questions, and the base model has a decent probability of solving 9 of them and a low probability of solving the remaining one. If one naively applies GRPO, then due to “transfer of diversity degradation”, the model might end up solving 9 questions with probability 1 and solving the remaining 1 question with probability 0. This gives pass@1 = 0.9. With OBE, the model maintains exploratory ability and might be able to explore the correct answer of the remaining one as it avoids the transfer of diversity degradation, thus the model might only have 0.95 probability of solving the same 9 questions but e.g., a probability of 0.8 of solving the one that GRPO is unable to solve. This gives a pass@1 = 0.935, and the pass@k rate should always dominate GRPO. We hope this example can illustrate the reason behind OBE and we are happy to answer any further questions if the reviewer has any remaining concern.
> > > >
> > > >
> > > > [1] Xu Y, Dong H, Wang L, et al. Scalable chain of thoughts via elastic reasoning. arXiv preprint arXiv:2505.05315, 2025.
> > > >
> > > > [2] Chen, Mark, et al. "Evaluating large language models trained on code." *arXiv preprint arXiv:2107.03374* (2021).
> > > >
> > > > [3] Yue, Yang, et al. "Does reinforcement learning really incentivize reasoning capacity in llms beyond the base model?." arXiv preprint arXiv:2504.13837 (2025).
> > > >
> > > > [4] Zhao, Rosie, et al. "Echo chamber: Rl post-training amplifies behaviors learned in pretraining." arXiv preprint arXiv:2504.07912 (2025).
> > > >
> > > > [5] Wu, Fang, et al. "The Invisible Leash: Why RLVR May or May Not Escape Its Origin." arXiv preprint arXiv:2507.14843 (2025).
> > > >
> > > > [6] Brown, Bradley, et al. "Large language monkeys: Scaling inference compute with repeated sampling." *arXiv preprint arXiv:2407.21787* (2024).
> > > >
> > > > [7] Wu, Yangzhen, et al. "An empirical analysis of compute-optimal inference for problem-solving with language models." (2024).

---

### Official Review · Reviewer_xnKV · 2025-11-01

**Soundness:** 1
**Presentation:** 2
**Contribution:** 2
**Rating:** 2
**Confidence:** 4

**Summary:**

The paper studies why RLVR for reasoning LLMs often collapses generation diversity, hurting pass@k. Treating RL post-training as a sampling process, the authors identify (i) a transfer of diversity degradation from solved to unsolved problems and (ii) the tractability of the outcome space (few distinct answers per question). They introduce Outcome-Based Exploration (OBE): historical variants that add UCB-style bonuses on final answers—with mean or constant baselines (OBE-Mean/OBE-Con) to balance positive/negative signals—and a batch variant (OBE-Batch) that penalizes within-batch repetition to directly promote test-time diversity.

**Strengths:**

1. The paper cleanly frames the diversity-collapse issue in outcome-based RL and separates it into two phenomena, making the problem setup easy to follow.


2. The proposed OBE variants are simple to implement on top of existing RLVR pipelines, requiring only outcome-level bookkeeping and modest additional logic.

**Weaknesses:**

1. On the claim of “transfer of diversity degradation across questions”: prior work already shows that RL sharpens model distributions [1, 2, 3], so the novelty here is limited. Moreover, the explanation that degradation occurs “because the model does not update on questions it has not solved yet” is unconvincing—lack of per-instance gradient does not imply lack of indirect effects. If updates do not generalize across questions, why should we expect generalization to the test set? As written, this argument needs stronger support.


2. On the observation that “diversity is tractable on verifiable domains”: the use of final-answer multiplicity as a proxy for diversity on unsolved problems feels assumed rather than demonstrated. Please provide evidence or a stronger justification that outcome-level diversity correlates with trajectory-level diversity in the unsolved regime.


3. In Figure 4, gains are visible on Llama, but for Qwen the OBE variants appear to yield little or no improvement; a similar pattern holds in Figure 6.

**Questions:**

1. For the claim that “RL eventually solves fewer questions than the base model,” the methodology of defining k = n t (accumulating generations across training checkpoints) is unusual. Would it be more natural to compute standard Pass@k independently at each checkpoint? What is the motivation for cross-checkpoint accumulation, and how should we interpret it relative to the conventional metric?


2. What sampling temperatures are used pre- and post-RL? Since several works report that RL sharpens the distribution [1, 2, 3], fixing a single temperature can be unfair [2]. A fairer protocol would sweep temperature per model and report the best Pass@k (with seeds/error bars). Do the conclusions remain under this evaluation?


[1] ProRL: Prolonged Reinforcement Learning Expands Reasoning Boundaries in Large Language Models https://arxiv.org/html/2505.24864v1

[2] Decomposing Elements of Problem Solving: What "Math" Does RL Teach? https://arxiv.org/pdf/2502.17356v1

[3] Rewarding the Unlikely: Lifting GRPO Beyond Distribution Sharpening https://arxiv.org/abs/2506.02355

---

> ### Author Response · Authors · 2025-11-21
> **Author Rebuttal (1/2)**
>
> We appreciate the reviewer’s feedback. Below we address the reviewer’s comment and we look forward to a constructive discussion period.
>
> > On the claim of “transfer of diversity degradation across questions”: prior work already shows that RL sharpens model distributions [1, 2, 3], so the novelty here is limited. Moreover, the explanation that degradation occurs “because the model does not update on questions it has not solved yet” is unconvincing—lack of per-instance gradient does not imply lack of indirect effects.
>
> We agree that the observation that RL sharpens model distributions (i.e., a decrease in pass@k with large k) is a well-observed phenomenon. The contribution of our paper is that we analyze the training dynamics of RL training and identify that the sharpening phenomenon is caused by a “transfer of diversity degradation”. I.e, “transfer of diversity degradation” is the explanation of the previously observed phenomenon of “RL sharpens model distributions”.
>
> Regarding the comment on generalization, we believe this might be a misunderstanding of our argument. We do not claim that degradation occurs because “the model does not update on questions it has not solved yet”, but the transfer of diversity degradation happens exactly due to certain undesirable generalization, as mentioned by the reviewer as well. Our argument is that, if there is no such undesirable generalization, since GRPO does not explicitly update on the questions it has not solved yet, we should not observe sharpening. This implies that the reason for sharpening is due to the undesirable generalization, i.e., “transfer of diversity degradation across questions”.
>
> > Please provide evidence or a stronger justification that outcome-level diversity correlates with trajectory-level diversity in the unsolved regime.
>
> We run an additional correlation analysis measuring whether outcome-level diversity tracks reasoning-level diversity on unsolved questions. For each model dataset pair, we pick 10 unsolved questions, and subsample $N=100$ generations from these questions. For these samples, we estimated trajectory-level reasoning diversity using an LLM judge (GPT-4o-mini). Specifically, we perform pairwise comparisons to label whether two solutions follow the same or different reasoning strategies. This gave us a score $s_{t} \in [0,1]$ for each pair. We then define the answer difference $s_a$, which is 1 if the pair has the same answer, and 0 if the pair has different answers.
>
> We then computed the Spearman correlation between $s_t$ and $s_a$ for each of the 4 settings we have, where **we obtain a score of 0.88 with a standard deviation of 0.03, indicating very strong correlation between the diversity of the outcome corresponds to the diversity of trajectory**. Note that most mismatches between $s_t$ and $s_a$ are intra-problem, meaning that sometimes the same outcome can have slightly different trajectories. We detail the prompt in Appendix A.3.
>
> > In Figure 4, gains are visible on Llama, but for Qwen the OBE variants appear to yield little or no improvement; a similar pattern holds in Figure 6.
>
> In Figure 4, we have several variants of OBE, and certain variants of OBE (e.g., OBE, OBE-Mean) indeed have little or no improvement for Qwen. They are not the final algorithm that we propose, and this is expected and we provide explanations in, e.g., section 3.1.1 and section 3.2 (now highlighted in green). Our final algorithms, OBE-Con and OBE-Batch, consistently outperform GRPO and other exploration baseline in all settings, and we refer the reviewer to Table 3 and 4 in the appendix for a direct quantitative comparison.
>
> To further demonstrate the effectiveness of OBE, we provide additional results in reasoning tasks beyond math, where we perform an additional experiment on the general reasoning dataset webinstruct [4], which includes subjects such as physics, chemistry, biology, economics, etc.. We filter the dataset to exclude math-related data, and we use a subset of the remaining data as the test set. We observe that, on domains other than math, OBE can further outperform vanilla GRPO, especially on pass@k with large k. Here is the detailed comparison:
>
> | pass@k | 1 | 2 | 4 | 8 | 16 | 32 |
> | --- | --- | --- | --- | --- | --- | --- |
> | Qwen-2.5-7B-Base | 22.5 | 29.8 | 36.2 | 42.3 | 46.4 | 49..7 |
> | GRPO | 37.5 | 41.5 | 45.4 | 48.3 | 51.6 | 53.9 |
> | OBE-Con | 39.9 (+2.4) | 44.1 (+2.6) | 48.1 (+2.7) | 51.9 (+3.6) | 54.9 (+3.3) | 57.8 (+3.9) |
> | OBE-Batch | 39.6 (+2.1) | 44.9 (+3.4) | 48.8 (+3.4) | 52.6 (+4.3) | 55.9 (+4.3) | 58.9 (+5.0) |
>
> We believe this result further demonstrates the effectiveness and generalness of our method. We refer to Appendix A.1 in the revised submission for more details.
>
> In Figure 6, the comparison is between OBE-Mean and OBE-Batch, and there is no baseline for simplicity of presentation, as both variants consistently outperform the baselines.

---

> > ### Author Response · Authors · 2025-11-21
> > **Author Rebuttal (2/2)**
> >
> > > For the claim that “RL eventually solves fewer questions than the base model,” the methodology of defining k = n t (accumulating generations across training checkpoints) is unusual. Would it be more natural to compute standard Pass@k independently at each checkpoint? What is the motivation for cross-checkpoint accumulation, and how should we interpret it relative to the conventional metric?
> > >
> >
> > We want to clarify that our comparison, different from previous ones, is conducted on the training accuracy/diversity instead of on the test set. Analyzing the dynamics of training gives us an explanation of why sharpening/diversity collapse happens (section 2.1). Thus our accumulation over the training is very natural because they are the samples generated during RL training, without any additional sampling from certain checkpoints. Note that this is a stronger diversity degradation result compared to the previous results obtained on test accuracy/diversity, which computes standard Pass@k independently at each checkpoint on the test set.
> >
> > > What sampling temperatures are used pre- and post-RL? Since several works report that RL sharpens the distribution [1, 2, 3], fixing a single temperature can be unfair [2]. A fairer protocol would sweep temperature per model and report the best Pass@k (with seeds/error bars). Do the conclusions remain under this evaluation?
> > >
> >
> > The sampling temperature for the base model (pre-RL) is 1 and we do not compare with post-RL (i.e., additional sampling from checkpoints). Our paper compares pre-RL and RL directly, which distinguishes us from the previous studies. Our sampling temperature during RL training is also 1, which, in general, is the highest reasonable temperature for RL training. We focus on the comparison of pass@k when k is large, which benefits from having a higher temperature, but a temperature higher than 1 is unnatural, especially for RL training. Thus we believe the current setting is optimal for both pre-RL and during-RL. That said, we are happy to test at any other reasonable temperature if the reviewer deems it necessary.
> >
> > [4] Xueguang Ma, Qian Liu, Dongfu Jiang, Ge Zhang, Zejun Ma, and Wenhu Chen. General-Reasoner: Advancing llm reasoning across all domains. arXiv:2505.14652, 2025.

---

> > > ### Comment · Reviewer_xnKV · 2025-11-27
> > >
> > > Thank you for your detailed response! While some of my concerns have been addressed, others remain. Please see my comments below.
> > >
> > > **Weakness 1**: When recent papers argue that "RL sharpens distribution," they typically refer to the observed decrease in pass@k on both training and test data. In my view, this framework already encompasses the training dynamics you describe. Consider two cases: (1) If the model has not yet seen a training question (e.g., before epoch 1 of RL), then that training question effectively functions as a test question, and the existing "sharpening" literature predicts reduced diversity. (2) If the model has encountered the training question in multi-epoch RL, decreased diversity is even more expected due to memorization. Could you clarify what novel insights your analysis provides beyond this existing framework?
> > >
> > > **Weakness 3**: Thank you for the pointer to Figure 4. My concern is that the performance appears highly non-linear due to unstable training dynamics, with different methods winning at different points. This raises questions about the results in Table 3: How did you select the total training steps and which specific checkpoints to evaluate? What criteria guided these choices?
> > >
> > > **Question 1**: Please see my comments on Weakness 1 above.
> > >
> > > **Question 2**: Using T=1 may not be optimal for the base model. I suggest considering an alternative evaluation methodology: for each model checkpoint (pre-RL, post-RL, etc.), identify the optimal sampling temperature and report performance at that "best" temperature. This would provide a fairer comparison across methods.
> > >
> > >
> > > **Overall concern**: Most results are based on Llama-8B-Instruct and Qwen-7B-Base. Why use a base model that hasn't undergone supervised fine-tuning? The current standard pipeline applies instruction tuning prior to RLVR. Could you justify this choice?

---

> > > > ### Author Response · Authors · 2025-11-27
> > > > **Response (1/2)**
> > > >
> > > > Thank you for your reply. We address the reviewer’s comments below:
> > > >
> > > > > **Weakness 1**: When recent papers argue that "RL sharpens distribution," they typically refer to the observed decrease in pass@k on both training and test data. In my view, this framework already encompasses the training dynamics you describe. Consider two cases: (1) If the model has not yet seen a training question (e.g., before epoch 1 of RL), then that training question effectively functions as a test question, and the existing "sharpening" literature predicts reduced diversity. (2) If the model has encountered the training question in multi-epoch RL, decreased diversity is even more expected due to memorization. Could you clarify what novel insights your analysis provides beyond this existing framework?
> > > > >
> > > >
> > > > As we mentioned in the previous rebuttal, the fundamental difference is: **previous studies make an observation (diversity degradation), and our study provides an explanation why diversity degradation happens (by studying the training dynamics)**. Previous studies only compare the base model vs. checkpoints of RL training (which is usually the last checkpoint), over the test dataset, and thus it only observes the **results of the RL training**. In order to study the dynamics of RL training, we have to propose a new technique, RL-as-sampling, which treats RL as a sampling algorithm; and we compare base model vs. RL-as-sampling, instead of only sampling from a selected few RL checkpoints. We also remind the reviewer that pass@k is different from diff@k, the first one measures “effective diversity”, and the second one measures true diversity. As far as we can tell, previous works only measure pass@k but not diff@k.
> > > >
> > > > We hope the previous remarks can already elucidate the difference between our study and the previous ones, but to answer the reviewer’s question:
> > > >
> > > > (1) if we only concern training within in first epoch, then it corresponds to comparison up to k=16, and RL can have a higher diff@k (diversity), a phenomenon opposite to sharpening.
> > > >
> > > > (2) We agree that if the policy discovers a correct answer then it should memorize it during the training, and in this case, for this specific question, pass@k goes up for all k, but diff@k should go down. This is expected and already discussed in the paper. However, the key is to track the diversity (pass@k) for unsolved questions during the training, where no memorization happens but diveristy (diff@k) still goes down. To our best knowledge, tracking diff@k for unsolved questions during training is new and this reveals “transfer of diversity degradation”, which is the explanation of sharpening for both train and test time. We refer the reviewer to Section 2.1 for a detailed argument on how the results reveal the transfer of diversity degradation and how this transfer explains sharpening.
> > > >
> > > > That said, we will add a discussion with the referred papers, and further highlight the difference of our study in the final version.
> > > >
> > > > > **Weakness 3**: Thank you for the pointer to Figure 4. My concern is that the performance appears highly non-linear due to unstable training dynamics, with different methods winning at different points. This raises questions about the results in Table 3: How did you select the total training steps and which specific checkpoints to evaluate? What criteria guided these choices?
> > > > >
> > > >
> > > > The total training step is the longest number of steps we could afford to train under our compute constraint. We believe it is sufficiently long to support the validity of our results.
> > > >
> > > > The checkpoint in Table 3 is the best checkpoint per model per pass@k during the whole training trajectory.

---

> > > > > ### Author Response · Authors · 2025-11-27
> > > > > **Response (2/2)**
> > > > >
> > > > > > **Question 2**: Using T=1 may not be optimal for the base model. I suggest considering an alternative evaluation methodology: for each model checkpoint (pre-RL, post-RL, etc.), identify the optimal sampling temperature and report performance at that "best" temperature. This would provide a fairer comparison across methods.
> > > > > >
> > > > >
> > > > > We hope our response above can already partially address this question. In our experiment, we do not compare with post-RL so there is no notion of model checkpoint except the base model. We compare base model and RL-as-sampling, and the current temperature (1) for RL is already optimal for pass@k with large k. Although we agree that T=1 might not be optimal for the base model (for pass@k with large k), we believe additional experiments with high temperature are unnecessary for two reasons: 1. the base model already outperforms RL-as-sampling with a potentially suboptimal temperature. 2. Even if a higher temperature can further improve the base model under large k, results derived from a temperature higher than 1 do not have very meaningful practical implications.
> > > > >
> > > > > > **Overall concern**: Most results are based on Llama-8B-Instruct and Qwen-7B-Base. Why use a base model that hasn't undergone supervised fine-tuning? The current standard pipeline applies instruction tuning prior to RLVR. Could you justify this choice?
> > > > > >
> > > > >
> > > > > We think this exactly highlights the generality of our results: we demonstrate that OBE is effective on both instruct and base models. In general, most papers in a similar domain only test one model, and unless we missed something, two of the three papers that the reviewer refers to use a non-instruct model. That said, the Qwen-7B-Base model already has sufficient instruction following capability [4], and it is in fact a much more standard choice than the instruct counterpart, which already goes through significant math reasoning post training [5,6].
> > > > >
> > > > > [4] Qwen, A. Yang, et al. "Qwen2. 5 technical report." *arXiv preprint* (2024).
> > > > >
> > > > > [5] Wang, Yiping, et al. "Reinforcement learning for reasoning in large language models with one training example." *arXiv preprint arXiv:2504.20571* (2025).
> > > > >
> > > > > [6] Cheng, Daixuan, et al. "Reasoning with exploration: An entropy perspective." *arXiv preprint arXiv:2506.14758* (2025).

---

### Official Review · Reviewer_aYim · 2025-11-01

**Soundness:** 2
**Presentation:** 2
**Contribution:** 2
**Rating:** 2
**Confidence:** 4

**Summary:**

This work studies the RL training dynamics of reasoning LLMs in the math domain. It finds that RL reduces effective diversity even on the training set compared to the base model. Specifically, it identifies, (i) diversity transfer, where reduced diversity on solved problems carries over to unsolved ones, and (ii) limited outcome space, since math reasoning tasks allow only a few distinct correct answers..  To address this,  the authors propose outcome-based exploration (OBE), which assigns exploration bonuses based on final outcomes. OBE includes two variants: 1) historical exploration, rewards rarely seen answers using UCB-style bonuses, and 2) batch exploration, penalizes duplicate answers within a batch to encourage test-time diversity. Experiments on competitive math tasks with Llama 3.1-8B and  Qwen 2.5-7B models show that both methods improve accuracy and help prevent diversity collapse.

**Strengths:**

1) The observation of the transfer of diversity degradation is insightful and inspirational for the community.
2) The idea of adding UCB-based exploration bonuses based on final answers is direct and simple to implement.

**Weaknesses:**

1) The performance improvement of OBE according to Figure 1 is not significant in the math domain. Since the method is general, authors can consider trying it in more domains.
2) The novelty of applying UCB-based exploration bonuses into RL is not that high, given many existing work in Deep RL.

**Questions:**

1. Do authors have results for other domains to validate the effectiveness of the method?
2. Better explanation of Figure 2 maybe useful for readers.

---

> ### Author Response · Authors · 2025-11-21
> **Author Rebuttal**
>
> Thank you for your review. We address the reviewer’s comment below and we look forward to a constructive discussion period.
>
> > The performance improvement of OBE according to Figure 1 is not significant in the math domain.
> >
>
> According to Figure 1, our method achieves 21% → 24% (a **14% improvement over GRPO**) on pass@1, 38% → 44% (a **15.8% improvement over GRPO**) on pass@32 with LLaMa model, and 40% → 42.5% (a **6.25% improvement over GRPO**) on pass@1, 57% → 60% (a **5.2% improvement over GRPO**) on pass@32 with Qwen model. These results are aggregated over 4 test datasets and 3 random seeds. In addition, our improvement is pointwise over all pass@k. We believe these results are significant enough to demonstrate the effectiveness of our method, which other reviewers also acknowledge. Finally, we also include a comparison with another entropy-based exploration baseline [1], in Table 4, and OBE outperforms the baseline on both accuracy and diversity.
>
> [1] Daixuan Cheng, Shaohan Huang, Xuekai Zhu, Bo Dai, Wayne Xin Zhao, Zhenliang Zhang, and Furu Wei. Reasoning with exploration: An entropy perspective. arXiv preprint arXiv:2506.14758, 2025.
>
> > Since the method is general, authors can consider trying it in more domains.
> >
>
> We appreciate the suggestion. For the rebuttal, we performed an additional experiment on the general reasoning dataset webinstruct [2], which includes subjects such as physics, chemistry, biology, economics, etc. We filter the dataset to exclude math-related data, and we use a subset of the remaining data as the test set. We observe that, on domains other than math, OBE can further outperform vanilla GRPO, especially on pass@k with large k. Here is the detailed comparison:
>
> | pass@k | 1 | 2 | 4 | 8 | 16 | 32 |
> | --- | --- | --- | --- | --- | --- | --- |
> | Qwen-2.5-7B-Base | 22.5 | 29.8 | 36.2 | 42.3 | 46.4 | 49.7 |
> | GRPO | 37.5 | 41.5 | 45.4 | 48.3 | 51.6 | 53.9 |
> | OBE-Con | 39.9 (+2.4) | 44.1 (+2.6) | 48.1 (+2.7) | 51.9 (+3.6) | 54.9 (+3.3) | 57.8 (+3.9) |
> | OBE-Batch | 39.6 (+2.1) | 44.9 (+3.4) | 48.8 (+3.4) | 52.6 (+4.3) | 55.9 (+4.3) | 58.9 (+5.0) |
>
> We believe this result further demonstrates the effectiveness and generality of our method. We refer to Appendix A.1 in the revised submission for more details.
>
> [2] Xueguang Ma, Qian Liu, Dongfu Jiang, Ge Zhang, Zejun Ma, and Wenhu Chen. General-Reasoner:
> Advancing llm reasoning across all domains. arXiv:2505.14652, 2025.
>
> > The novelty of applying UCB-based exploration bonuses into RL is not that high, given many existing work in Deep RL.
> >
>
> We want to clarify that OBE is, in fact, different from the traditional count-based exploration (for example, using UCB in deep RL). In traditional (deep) RL/bandit, UCB keeps track of the count of each individual state and action, but naively applying state-based UCB is intractable in the LLM reasoning setting because we have an exponential number of states in LLM reasoning. This exactly motivates the proposed outcome-base exploration, where we only perform exploration in the outcome space, which is tractable. In fact, this setting is even little studied in the bandit setting, let alone deep RL. We refer the reviewer to Section 4.2 of the paper for the distinction of OBE from traditional UCB-based exploration and the theoretical benefit of OBE.
>
> > Better explanation of Figure 2 maybe useful for readers.
> >
>
> We revise the caption of Figure 2 in blue and highlight other explanation texts in the revised submission. Please let us know if there is any specific improvement we can make beyond the revised version.

---

### Author Response · Authors · 2025-11-21
**General Response**

We thank the reviewers for their time. In this section, we summarize the additional experiments we performed for the rebuttal, and we believe the new results further demonstrate the validity and effectiveness of our proposed algorithms. We present these new results in Appendix A.

- In Appendix A.1, we perform additional experiments on settings **beyond math reasoning**. We train on a subset of the webinstruct dataset, which include reasoning questions over domains such as physics, chemistry, biology, etc. We exclude the questions within the math domain. We train the Qwen-2.5-7B-Base model and we observe **our proposed algorithm, OBE-Con and OBE-Batch, significantly outperform the GRPO baseline**, especially for pass@k with large k.
- In Appendix A.2, we perform ablations on the choice of the bonus coefficient and baseline in OBE-Con and OBE-Batch. The ablation shows a natural tradeoff between diversity and correctness by different scales of the hyperparameters.
- In Appendix A.3, we conduct an analysis measuring whether outcome-level diversity tracks reasoning-level diversity on unsolved questions. With LLM-as-a-judge, we show significant statistical evidence that there is a strong correlation between the diversity of the outcome to the diversity of the trajectory.

---

### Meta-Review · Area_Chair_ekm5 · 2025-12-10

**Summary:**

The common issues lied in both the methodological perspective on the exploration scheme, and the experiment perspective on limited scope.

Most reviewers replied to the rebuttals, however, the authors failed to convince the reviewers to adjust their scores. There might exist some different opinions. The authors were suggested to improve their work carefully.

So it has to be rejected.

**Reviewer Concerns:**

The concern from Reviewer UQnj was addressed, however, other reviewers remained.

**Reviewer Scores:**

It received 4 reviews with the average score 3.5.

Nobody.

---

### Decision · Program_Chairs · 2026-01-26

Reject